

# Dynamic changes in outlet glaciers in northern Greenland from 1948 to 2015

Emily A. Hill[1], J. Rachel Carr[1], Chris R. Stokes[2], G. Hilmar Gudmundsson[3]

[1]School of Geography, Politics, and Sociology, Newcastle University, Newcastle-upon-Tyne, NE1 7RU, UK
[2]Department of Geography, Durham University, Durham, DH1 3TQ, UK
[3]Department of Geography and Environmental Sciences, Northumbria University, Newcastle-upon-Tyne, NE1 8ST, UK

*Correspondence to*: Emily A. Hill (e.hill3@newcastle.ac.uk)

## Abstract

The Greenland Ice Sheet (GrIS) is losing mass in response to recent climatic and oceanic warming. Since the mid-1990s,
marine-terminating outlet glaciers across the GrIS have retreated, accelerated and thinned, but recent changes in northern Greenland have been comparatively understudied. Consequently, the dynamic response (i.e. changes in surface elevation and velocity) of these outlet glaciers to changes at their termini, particularly calving from floating ice tongues, remains unknown. Here we use satellite imagery and historical maps to produce an unprecedented 68-year record of terminus change across 18 major outlet glaciers and combine this with previously published surface elevation and velocity datasets. Overall, recent (1995–
2015) retreat rates were higher than at any time in the previous 47 years, but change-point analysis reveals three categories of frontal position change: (i) minimal change followed by steady and continuous retreat, (ii) minimal change followed by a switch to a period of short-lived rapid retreat, (iii) glaciers that underwent cycles of advance and retreat. Furthermore, these categories appear to be linked to the terminus type, with those in category (i) having grounded termini and those in category (ii) characterised by floating ice tongues. We interpret glaciers in category (iii) as surge-type. Glacier geometry (e.g. fjord
width and basal topography) is also an important influence on the dynamic re-adjustment of glaciers to changes at their termini. Taken together, the loss of several ice tongues and the recent acceleration in the retreat of numerous marine-terminating glaciers suggests northern Greenland is undergoing rapid change and could soon impact on some large catchments that have capacity to contribute an important component to sea level rise.

## 1 Introduction

Mass loss from the Greenland Ice Sheet (GrIS) has accelerated since the early 2000s, compared to the 1970s and 80s (Kjeldsen et al., 2015; Rignot et al., 2008), and could contribute 0.45–0.82 m of sea level rise by the end of the 21[st] century (Church et al., 2013). Recent mass loss has been attributed to both a negative surface mass balance and increased ice discharge from marine-terminating outlet glaciers (van den Broeke et al., 2016; Enderlin et al., 2014). The latter contributed ~40% of total mass loss across the GrIS since 1991 (van den Broeke et al., 2016), and increased mass loss was synchronous with widespread



glacier acceleration from 1996 to 2010 (Carr et al., 2017b; Joughin et al., 2010; Moon et al., 2012; Rignot and Kanagaratnam, 2006). Coincident with glacier acceleration, dynamic thinning at elevations of <2000 m elevation has occurred on fast flowing marine-terminating outlet glaciers (Abdalati et al., 2001; Krabill et al., 2000), particularly in the south-east and north-west regions of the ice sheet (Moon et al., 2012; Pritchard et al., 2009). Alongside thinning and acceleration, terminus retreat has

been widespread since the 1990s across the ice sheet (e.g. Box and Decker, 2011; Carr et al., 2017b; Jensen et al., 2016; Moon and Joughin, 2008) and several studies have identified terminus retreat as a key control on inland ice flow acceleration and dynamic surface thinning (Howat et al., 2005; Joughin et al., 2004, 2010; Nick et al., 2009; Thomas, 2004; Vieli and Nick, 2011).

Most previous work has concentrated on central-west and south-east Greenland, and most notably at Jakobshavn Isbræ, Helheim, and Kangerdlugssuaq Glaciers (e.g. Joughin et al., 2004; Howat et al., 2005; Howat et al., 2007; Nick et al., 2009). Observations at all three glaciers showed acceleration and surface thinning following terminus retreat and, at Jakobshavn, this was in response to the collapse of its floating ice tongue (Amundson et al., 2010; Joughin et al., 2008; Krabill et al., 2004). In both regions atmospheric/ocean warming and the retreat of sea-ice have been hypothesised as the main triggers of recent 21[st]

century glacier retreat (e.g. Bevan et al., 2012; Cook et al., 2014a; Holland et al., 2008; McFadden et al., 2011; Moon and Joughin, 2008). However, variability in the magnitude and rate of retreat between glaciers occurs due to local factors, such as the width and depth of fjords (Carr et al., 2013; Enderlin et al., 2013; Howat et al., 2007; Porter et al., 2014).

In the northern Greenland, several glaciers have thinned (Rignot et al., 1997), accelerated (Joughin et al., 2010a), and retreated,

and have lost large sections of their floating ice tongues between 1990 to 2010 (Box and Decker, 2011; Carr et al., 2017b; Jensen et al., 2016; Moon and Joughin, 2008; Murray et al., 2015). This region is also characterised by large fjord-terminating outlet glaciers, many of which terminate in kilometres-long floating ice tongues, while several others are potentially surge-type (Hill et al., 2017; Joughin et al., 1996; Reeh et al., 2003; Rignot et al., 2001). However, far fewer studies have focussed on northern Greenland, except for at Petermann and the Northeast Greenland Ice Stream (NEGIS) (e.g. Khan et al., 2015; Nick

et al., 2012), and there are few datasets of longer-term changes at glacier fronts and their potential impact on ice flow further inland.

Here we present changes in frontal position, ice velocity and surface elevation over the last 68 years (1948 to 2015) in northern Greenland. We evaluate the dynamic response of glaciers in the region to observed changes at the terminus, focusing on the

influence of the presence or absence of floating ice tongues. Our study region includes 18 major marine-terminating outlet glaciers (Figure 1), which are the main outlets in the region, and together drain approximately 40% of the GrIS by area (Hill et al., 2017; Rignot and Kanagaratnam, 2006). First, we provide a multi-decadal record of annual terminus positions between 1948 and 2015 to assess long-term change. These data are then used to objectively categorise different types of terminus behaviour, which appear to be linked to different types of glacier (e.g. those with grounded termini, those with floating termini



and those which may be surge-type). Finally, we compile recently published datasets of surface elevation and ice velocity, to investigate the dynamic response of these categories of glaciers to frontal position change.

## 2 Methods

### 2.1 Terminus change

**2.1.1 Data sources**

Terminus positions of 18 study glaciers in northern Greenland (Figure 1) were manually digitised from a combination of satellite imagery and historical topographic navigational charts between 1948 and 2015 (Table S1). From 1975 to 2015 we used Landsat 1–5 MSS (1975–1994), Landsat 7 TM (2000–2013) and Landsat 8 (2013–2015). These scenes were acquired from the United States Geological Survey (USGS) Earth Explorer website (earthexplorer.usgs.gov). To reduce the influence

of seasonal changes in terminus position, scenes were selected from late summer each year, and 70% were within one month of the 31st August. Several Landsat MSS images required additional georeferencing and were georeferenced to 2015 Landsat 8 images. Early Landsat scenes (1970–1980s) were supplemented with SPOT–1 imagery from the European Space Agency (ESA) (intelligence-airbusds.com). These scenes covered 8 of 18 study glaciers in 1986/87, and were also selected from late August. SPOT–1 scenes were also georeferenced to 2015 Landsat Imagery. Additionally, we used aerial photographs (2 m

resolution), which were provided orthorectified by Korsgaard et al. (2016). These covered all study glaciers between Humboldt east to L. Bistrup Bræ in 1978, and Harald Moltke Bræ, Heilprin and Tracy Glaciers in NW Greenland in 1985 (Korsgaard et al., 2016).

To extend the record of glacier terminus positions further back in time, declassified spy images from the Corona satellite were

acquired from the USGS Earth Explorer website (Table S1), which covered 5 of 18 glaciers in 1962/63 and Petermann and Ryder Glaciers in 1966. These images were georeferenced to a Landsat 8 scene from 2015 imagery with total RMSE errors of 105 to 360 m. Frontal position changes smaller than this were discounted from the assessment. To further assess the historical terminus positions of the glaciers we used navigational map charts from the United States Air Force 1:1,000,000 Operational Navigation Charts from 1968/69 (lib.utexas.edu/maps/onc/). These were made available through the Perry-Castañeda Library,

courtesy of the University of Texas Libraries, Austin. Data from 1948 comes from AMS C501 Greenland 1: 250,000 Topographic Series maps distributed by the Polar Geospatial Centre (pgc.umn.edu/data/maps/). All maps were georeferenced to 2015 Landsat imagery using a minimum of 10 ground control points (GCPs), which were tied to recognisable stationary features such as on nunataks and fjord walls. RMSE errors across all glaciers ranged between 150 and 510 m.



### 2.1.2 Front position mapping

Changes in glacier frontal positions were measured using the commonly adopted box method, which accounts for uneven calving front retreat (e.g. Carr et al., 2013; Howat and Eddy, 2012; Moon and Joughin, 2008). For each glacier, a rectilinear box was drawn parallel to the direction of glacier flow (Figure 2), and extending further inland than the minimum frontal

position. Due to Steensby Glacier's sinuous fjord, a curvilinear box was used (see Lea et al., 2014). Glacier frontal positions were digitised in sequential images and the difference between successive terminus polygons give area changes over time within the box. Dividing these areas by the width of the reference box derives width-averaged relative glacier front positions.

Aside from georeferencing errors outlined in the previous section, the main source of error associated with frontal positions is

attributed to manual digitisation (e.g. Carr et al., 2013; Howat and Eddy, 2012; Moon and Joughin, 2008). We quantified this by repeatedly digitising a ~3 km section of rock coastline 20 times for each image type or map source. The resultant total mean errors were: 3.6 m for Landsat 8, 19 m for Landsat 7 ETM, 17 m for Landsat MSS, 20 m for SPOT–1, 16 m for Orthophotographs, 21 m for Corona, and 27 m for historical maps. Overall, the mean total error primarily associated with manual digitising was 19 m, which is below the pixel resolution of all imagery sources except the 15-m panchromatic Landsat

band. Similar to other studies (e.g. Bevan et al., 2012; Howat and Eddy, 2012; Murray et al., 2015), an additional source of error occurs in selecting the correct terminus position at several glaciers, due to the presence of year-round sea ice and the fractured nature of the glacier terminus. This was particularly the case at Steensby and C. H. Ostenfeld and glaciers draining the Northeast Greenland Ice Stream (Figure 1). These glaciers were the only ones affected by these uncertainties, and so we re-digitised all Landsat terminus positions from 1999 to 2015 to estimate the error in mapping the terminus position. Additional

errors were calculated to be ± 13 % for Steensby, C. H. Ostenfeld, Nioghalvfjerdsfjorden, and Zachariae Isstrøm. At these glaciers, similar inaccuracies in identifying the true glacier terminus may have occurred by the authors of the earliest map charts (1948 and 1969), and we therefore consider these to be a broad estimate of the past location of glacier termini rather than exact frontal positions.

### 2.1.3 Changepoint analysis

We used 'changepoint' analysis to objectively test whether different categories of terminus change behaviour exist in northern Greenland. Changepoint analysis is used to identify significant breaks in time-series data, and has previously been used to identify changes in the terminus behaviour of outlet glaciers (e.g. Carr et al., 2017a; Bunce et al., *in review*). Here, we employ a similar technique to detect statistically significant breaks in frontal position data across 18 outlet glaciers in northern Greenland. To do this we use the 'findchangepts' function in MATLAB software which employs the methodology of Killick

et al. (2012) and Lavielle (2005) used in similar packages in R software. Linear regression was used to detect significant breaks in the normalised frontal position time series, based on the mean and coefficients of the linear regression equation line slope and intercept either side of a change point. Similar to previous studies, we set the minimum distance between points to 4 (Carr





et al., 2017a), to only allow breaks >4 years to occur. This number must be small enough to allow for breaks not to be missed, but also large enough so that breaks do not incorrectly occur between every data point. The results are highly insensitive to incrementing the number up and down within this range. We also include a minimum threshold penalty value using the mean terminus position, which only allows for a changepoint when total error decreases by the minimum threshold. This penalty

value then allows for automatic estimation of the number of change points along each time series of frontal position data.

## 2.2 Ice velocity and surface elevation

Previously published datasets of annual ice velocity and surface elevation change were compiled to assess dynamic glacier changes in northern Greenland. Velocity and surface elevation change datasets are generally only available from 1990 onwards. The earliest velocity maps from winters 1991/92 and 1995/96 were acquired from the European Remote Sensing (ERS)

satellites (1 and 2), as part of the ESA Greenland Ice Sheet CCI (Climate Change Initiative) project (Nagler et al., 2016). The first (1991/92) covers northern Greenland drainage basins from Humboldt and then east to Hagen Bræ, and the second (1995/96) covers all 18 study glaciers. We estimated average errors in velocity magnitude across all northern Greenland drainage basins, which were 2.5 m a$^{-1}$ for 1991/92 and 10 m a$^{-1}$ for 1995/96.

Subsequent velocity datasets were primarily acquired from the NASA MEaSUREs program (Joughin et al., 2010). These velocity maps were derived from 500 m resolution Interferometric Synthetic Aperture Radar (InSAR) pairs from the RADARSAT satellite in winter 2000/01, and then annually from winter 2005/06 to 2009/10 (Joughin et al., 2010). To assess velocity errors, we use the published error estimates to calculate mean velocity errors across all years and spanning all drainage catchments in northern Greenland. Mean velocity errors were 6.3 m a$^{-1}$. For 7 study glaciers, additional annual velocity data,

derived from ERS1, ERS2 and Envisat satellites, were available annually between 1991/92 to 1997/98 and between 2003/04 to 2009/10 from the ESA Greenland CCI project (Nagler et al., 2016). Winter velocities from these data were calculated from October to April.

For the winters of 2010/11, 2011/12 and 2012/13, glacier velocity maps were also derived from InSAR (TerraSAR-X image

pairs) for 11 of 18 study glaciers (Joughin et al., 2011). Despite higher spatial resolution (100 m), these maps are limited to the grounding line and extend 27–56 km inland. Mean error for these data is 23 m a$^{-1}$ across all years. Winter velocities for 2013/14 were derived from intensity tracking of RADARSAT-2 satellite data, and from offset tracking of Sentinel-1 radar data for 2014/15 and 2015/16, as part of the ESA CCI project (Nagler et al., 2016). The mean error of these data from a central section of northern Greenland is 7.3 m a$^{-1}$ (Nagler et al., 2015). Using the earliest full regional velocity map (1995/96) and the

most recent record (2015/16), the rate of annual velocity change was calculated over this 20-year period.

We use surface elevation change (SEC) data from ERS-1, ERS-2, Envisat, and Cryosat-2 radar altimetry for 1992 to 2015, and made available by the ESA's Greenland Ice Sheet CCI project (Khvorostovsky, 2012; Simonsen and Sørensen, 2017; Sørensen



et al., 2015). Data from 1992 to 2011 were derived from the ERS-1, ERS-2 and Envisat satellites, using a combination of cross-over and repeat track analysis, which have then been merged to create a continuous dataset across satellites (Khvorostovsky, 2012). These data are provided in 5-year running means from 1992 to 2011 and at a resolution of 5 km. For the most recent SEC (2011 to 2015), we used Crysosat-2 satellite elevation change which are provided in 2-year means

(Simonsen and Sørensen, 2017). These data were generated using the Least Mean Squares method, where grid cells were subtracted from the Greenland Ice Mapping Project (GIMP) DEM (Howat et al., 2014) and corrected for backscatter and leading edge width (Simonsen and Sørensen, 2017). Calculations were made at a 1 km grid resolution and resampled to 5 km to conform with 1992–2011 datasets (Simonsen and Sørensen, 2017). Using error estimates (Simonsen and Sørensen, 2017), we calculated mean errors across all years and across all northern Greenland drainage basins to be ±0.14 m a$^{-1}$. To compare

changes in SEC from 1992 to 2015, 5-year running means (m a$^{-1}$) from 1992 to 1996 were differenced with the most recent estimates of SEC for the two-year period 2014–2015.

Profile data from all velocity and surface elevation time series were extracted along each glacier centreline, which were drawn following the method of Lea et al. (2014). The Euclidean distance was calculated between parallel fjord walls that were

digitised in 2015 Landsat 8 imagery. The maximum distance line was then traced from the furthest terminus extent back to the ice divide. Annual profiles of velocity and SEC were sampled at 500 m along each glacier centreline (Figure 2).

## 2.3 Fjord width, ice surface and basal topography

Fjord width was measured perpendicular to glacier centrelines following the method of Carr et al. (2014). Points were extracted at 500 m intervals along each fjord wall and joined by lines that crossed the fjord. The length of these lines is the width between

the fjord walls, and changes along each fjord were fitted with a linear regression model to determine if the fjord widens or narrows with distance inland. To determine the bathymetry of each study glacier in northern Greenland, regional ice surface topography was acquired from the GIMP DEM, with a resolution of 150 m (Howat et al., 2014). Ice thickness and bed topography data were taken from the Operation Ice Bridge BedMachine v2 dataset, which uses radar ice thicknesses data, and the mass conservation method, to derive ice thicknesses (Morlighem et al., 2014). Bed topography was then derived from

subtracting ice thickness from the GIMP surface DEM (Morlighem et al., 2014). Surface, ice thickness and bed topography were also extracted at 500 m intervals along glacier centrelines. Bed profiles were then fit with a linear regression model to establish retrograde or seaward sloping beds. To estimate drainage catchment areas and the percentage of each catchment below present sea level for each study glacier, surface drainage catchments were delineated using the GIMP surface DEM and topographic analysis functions within TopoToolbox in MATLAB (Schwanghart and Kuhn, 2010). First, all sinks in the DEM

were filled and the resultant DEM was used calculate flow direction and flow accumulation. Secondly, we used a flow accumulation threshold of 500 to calculate stream order. Flow direction and stream order gridded outputs were then used to delineate surface drainage catchments.





## 2.4 Climatic and oceanic data

To provide some ocean-climate context to the observations of each outlet glacier, we acquired data on air temperatures and sea ice concentrations. It is beyond the scope of this paper to undertake a detailed assessment of the precise drivers of recent retreat at each glacier and we instead focus on dynamic glacier change in response to terminus perturbations. This is in part due to the limited availability of detailed region wide climate-ocean data for northern Greenland. Annual surface air temperatures were taken from the only two long-term automatic weather stations in the region: Pituffik (PK) (76°32'N, 68°45'W) in northwest Greenland, and Danmarkshavn (DK) (76°46'N, 18°40'W), in northeast Greenland (Figure 1). These datasets are provided by the Danish Meteorological Institute (DMI) as part of the historical climate data collection (Vinther et al., 2006), and were chosen due to their long and continuous record from 1948–49 to 2015. Mean air temperatures were subjected to changepoint analysis to highlight potentially significant breaks in annual air temperature time series data during the study period. Daily air temperatures from these stations which cover 1974 to 2006 (PK) and 1958 to 2013 (DK) were also used to calculate the number of positive degree days (PDDs), and mean summer (June, July August) temperatures.

Sea ice concentrations (SIC) from the Nimbus-7 SMMR, and Special Sensor Microwave/Imager (SSM/IS) sensors from the Defence Meteorological Satellite Program (DMSP), were acquired from the National Snow and Ice Data Centre (NSIDC: Cavalieri et al., 1996). These data provide the longest continuous record (1979–2015) of sea ice conditions across northern Greenland, although at the expense of a relatively coarse resolution (25 km), which may compromise the accuracy of SICs near the glacier terminus. Thus, these can only be used to assess region wide changes in sea ice conditions. Annual SICs were taken from September each year, which is considered the annual minimum of sea ice extent. These September estimates were split into decadal anomalies from the 1979 to 2015 mean. Due to the absence of accurate and systematic regional datasets of both sea surface and subsurface temperatures, we do not assess the regional impacts of ocean temperatures on dynamic outlet glacier change in northern Greenland.

## 3. Results

### 3.1 Changes in glacier frontal position (1948–2015)

Across northern Greenland, 13 of the 18 study glaciers underwent overall retreat between 1948 and 2015, while the remaining five advanced (Figure 3). Long-term glacier retreat rates (1948–2015) ranged between -15 m a$^{-1}$ at Marie-Sophie Glacier, to twenty times greater at Petermann Glacier (-311 m a$^{-1}$). At Petermann Glacier, the large retreat rate resulted from two large calving events in 2010 and 2012, which together removed 27 km of its floating ice tongue (Falkner et al., 2011; Johannessen et al., 2013). Zachariae Isstrøm, which partially drains the NEGIS, has a similarly high retreat rate of -282 m a$^{-1}$, which occurred steadily and culminated in the eventual loss of its 21-km floating ice tongue between 2002 and 2012 (Table 1). There was clear variability in the long-term overall retreat rates across northern Greenland. A further five glaciers had retreat rates that exceeded



-100 m a$^{-1}$ (Table 1), and the remaining 6 glaciers that underwent retreat did so at rates of -15 to -58 m a$^{-1}$. Between 1948 and 2015, Ryder, Storstrømmen and L. Bistrup Bræ Glaciers advanced at a similar rate (~40 m a$^{-1}$), while Brikkerne Glacier advanced at 82 m a$^{-1}$ (Table 1). Steensby Glacier underwent minimal change during the study period (1 m a$^{-1}$: 1948–2015), but with a high rate of retreat from 1978 to 2015 (-366 m a$^{-1}$).

While substantial terminus retreat has taken place at many glaciers in the region, there are large differences in the timing and magnitude of retreat between glaciers, and throughout the study period (Table 1, Fig.3). To assess the variability of retreat rates across northern Greenland, we present mean retreat rates across five decadal time periods (1948–1975, 1976–1985, 1986–1995, 1996–2005, 2006–2015) in Figure 4 (a-e), except for the earliest epoch which spans 27 years due to image availability.

During the first epoch (1948 to 1975) small advances and retreats took place across the region (< 500 m a$^{-1}$ magnitude). This was followed by a decade dominated by glacier advance (with some minor retreat at certain glaciers) between 1976 and 1985. Several glaciers with overall high retreat rates (e.g. Hagen Bræ, Zachariae Isstrøm, Petermann) underwent advance during this period. In the subsequent epoch (1986 to 1995), a mixture of advance and retreat occurred. Rates of frontal position change were greater than during previous intervals, ranging from -780 m a$^{-1}$ retreat at C. H. Ostenfeld to 750 m a$^{-1}$ advance at

Storstrømmen (Figure 4c). During the last two decades of the study period (1996 to 2015), retreat rates were substantially higher than in the previous three epochs, peaking at Petermann Glacier (-2200 m a$^{-1}$; Figure 4e, f). Retreat during this period at Hagen Bræ, Zachariae Isstrøm, Petermann far outweighed earlier advances.

## 3.2 Categories of terminus change behaviour

Changepoint analysis allowed us to objectively identify significant breaks in frontal position time series for each glacier. This

revealed three broad categories of frontal position change (Figure 5) in northern Greenland: 1) minimal frontal position change, followed by a period of steady and continuous glacier retreat after the change point (Fig.5a), 2) minimal frontal position change, followed by a switch to a period of short-lived rapid retreat (Fig.5b), 3) glaciers which experienced sustained periods of glacier advance (Fig.5c). Of note, is that the first category encompassed all outlet glaciers with grounded termini, and the second encompasses those with floating ice-tongues. To further test the veracity these categories of terminus behaviour, we use a two-

paired t-test between glacier retreat rates during these respective periods of steady or rapid retreat (Categories 1 and 2). Retreat rates are significantly different to a 99% confidence level (p-value 0.009), supporting these being distinct categories of terminus behaviour. Frontal position change rates in Category 3 (advancing glaciers), are also statistically different (99% confidence, p-value 0.004) from both categories 1 and 2. The following subsections cover the characteristics of frontal position change for these three types of outlet glacier in the region.

### 3.2.1 Category 1: Minimal change, followed by steady retreat

Eight of 18 study glaciers in northern Greenland fall into the first category of terminus behaviour, and we note that all of these are grounded at their terminus and lack floating ice tongues (Figure 7). For these glaciers, there was a transition from minimal



frontal position retreat/advance (e.g. Academy Glacier, Fig. 7o), to a period of steady retreat which lasted for an average of 26 years (Figure 6b). During the initial period of minimal change, frontal position change averaged -67 m a$^{-1}$ across these eight glaciers, which increased to -150 m a$^{-1}$ during the period of steady retreat (Figure 6b). Net retreat during this latter period ranged from -0.6 to 8 km, and the greatest total terminus changes took place at Tracy Glacier (8 km retreat: 1981–2015),

Harald Moltke Bræ (5 km retreat: 1988–2015), and Kofoed-Hansen Bræ (4.6 km: 1973–2015). The timing of this switch from minimal change to steady retreat was not uniform, but the majority of glaciers began steadily retreating from the 1990s to 2000s and continued at the same rate thereafter (Figure 6b).

### 3.2.2 Category 2: Minimal change, followed by rapid, short-lived retreat

The second category of glacier frontal position change encompasses six glaciers which currently, or recently, terminated in

long floating ice tongues. These glaciers also showed minimal terminus change at the beginning of the record, followed by short-lived rapid retreat, lasting <6 years on average (Figure 6a). During the phases of rapid retreat, rates ranged between -700 m a$^{-1}$ at Nioghalvfjerdsfjorden to -8997 m a$^{-1}$ at Petermann Glacier (Figure 6a), and were on average 40 times greater (-4536 m a$^{-1}$) than during the steady retreat phases at Category 1 glaciers. Rapid retreat was often followed by another period of minimal terminus change (e.g. Petermann Glacier and Hagen Bræ: Figure 6a). For five glaciers (Zachariae Isstrøm, Petermann,

Steensby, C. H. Ostenfeld, and Hagen Bræ), rapid retreat removed substantial floating ice sections (11.6–26 km net retreat: Figure 8), through large episodic calving events. This led to complete ice tongue loss at Zachariae Isstrøm by 2011/12, and at C. H. Ostenfeld, Steensby and Hagen Bræ by 2016 (Figure 8). Similar to Category 1 glaciers, the timing of the switch to rapid retreat is not synchronous, but mainly occurs after 1990 (Figure 6a). At most glaciers, the duration of rapid retreat was short-lived (< 5 years) in comparison to the duration of steady retreat (> 13 years) at Category 1 glaciers.

### 3.2.3 Category 3: Sustained periods of glacier advance

The final category of glacier terminus behaviour includes those which have shown cyclic periods of sustained glacier advance at some (or several) point(s) between 1948 and 2015 (> ~90 m a$^{-1}$: Table 1). Periods of terminus advance averaged ~ 420 m a$^{-1}$ and lasted for an average duration of 18 years (Figure 6c). All four glaciers began advancing in the 1970s (Figure 6c). Some adjacent glaciers (e.g. Storstrømmen and L. Bistrup Bræ) continued to advance for a similar period (~13–17 years from 1973

to 1990), before undergoing relatively limited terminus change from 2000 onwards (Figure 6c). Despite synchronous advance, their advance rates differed by almost an order of magnitude (89 m a$^{-1}$ at L. Bistrup Bræ, and 725 m a$^{-1}$ at Storstrømmen, Fig. 6c). At Ryder Glacier, there were four main cycles of glacier advance and retreat during the record. These took place between 1948–1996, 1968–1986, 1999–2006, and 2008–2015 and advance rates ranged from 183 to 750 m a$^{-1}$ (Figure 6c). Periods of advance were separated by generally shorter periods (2–13 years) of higher magnitude retreat (ranging from -960 to -1950 m

a$^{-1}$) (Figures 6c). Brikkerne Glacier showed 9 km advance between 1968 and 1978, but between 2000 and 2010 it showed very little change in front position (Figure 9a).





### 3.3 Ice velocity

#### 3.3.1 Category 1: Minimal change, followed by steady retreat

Most Category 1 outlet glaciers with grounded termini (6 of 8) accelerated along their centreline profiles (ranging from 0.32 to 37 m a$^{-1}$) from 1996 to 2016 (Table 1). Terminus acceleration equated to >27%, following the onset of steady retreat from

the 1990s. For example, Tracy and Heilprin accelerated substantially during their steady retreat periods (Table 1). At Heilprin Glacier this resulted in a 45% increase (607 to 878 m a$^{-1}$) in velocity from 2001 to 2016 (Figure 7a), during which the glacier retreated at -110 m a$^{-1}$ (Figure 6). Substantially greater acceleration took place at Tracy Glacier from 1996 to 2016 (156%), which was associated with higher magnitude retreat rates (-263 m a$^{-1}$: Figure 6). It was also clear that from 1996 to 2016, velocity increases propagated inland (~20 km) at both glaciers (Figure 7). Humboldt, Harder, and Marie-Sophie Glaciers

flowed more slowly than other grounded-terminus glaciers (< 400 m a$^{-1}$), but still showed large accelerations (27–108%) at their termini during steady retreat (Figure 6). Harald Moltke Bræ also accelerated between 1990 and 2016 (22 m a$^{-1}$: Table 1), and retreated at -196 m a$^{-1}$ (Figure 6b). However, it underwent two very large velocity increases (> 1000 m a$^{-1}$) between 2001 and 2006 and again during winter 2013/14, both of which coincided with short-lived glacier advance (0.5–0.8 km). Some grounded-terminus outlet glaciers did not show substantial acceleration following retreat: Academy Glacier and Kofoed-

Hansen Bræ had sustained periods of steady retreat, but showed distinct variability in velocity change throughout (Figure 7c,d), which cannot be linked to the timing of increased retreat rates (Figure 6).

#### 3.3.2 Category 2: Minimal change, followed by rapid, short-lived retreat

In contrast to Category 1 glaciers grounded at their terminus, most glaciers in Category 2 with floating ice tongues (5 of 6) showed minimal increases in velocity between 1996 and 2016 (Table 1). Following periods of rapid retreat, there were two

dominant patterns in velocity change: 1) several glaciers showed minimal (< 5%) increases in velocity, 2) other glaciers had short-lived acceleration that did not propagate inland. For example, Petermann and C. H. Ostenfeld only accelerated by <4% at their grounding line following the 26 and 19 km loss of their floating ice tongues, respectively (Figure 8b,i). Despite showing minimal acceleration, increased ice flow continued at these glaciers for several years after ice tongue loss (Figure 8b,i). Conversely ice tongues retreats of 14 and 12 km at Hagen Bræ and Steensby Glacier, were followed by greater but short-lived,

grounding line acceleration (< 13%) in the following winters (Figure 8a,c). Once both glaciers returned to periods of minimal terminus change, they subsequently decelerated (Figure 8). However, minimal/and or short-lived acceleration was not ubiquitous. Instead, ice tongue retreats for ~10 years at both glaciers draining the NEGIS (Figure 6) were synchronous with gradual glacier acceleration in the following decade (2006 to 2016: 43% at Zachariae Isstrøm and 10% at Nioghalvfjerdsfjorden). This prolonged, high magnitude glacier acceleration following retreat, is more characteristic of

Category 1 glaciers with grounded termini. Another key difference is that entire ice tongue removal at some glaciers (e.g. Zachariae Isstrøm in 2011/12) was succeeded by glacier acceleration (125 m a$^{-1}$: 2012 to 2016, Figure 8h), compared to



minimal acceleration following ice tongue removal at others (e.g. C. H. Ostenfeld and Hagen Bræ). Overall, most glaciers with floating ice tongues showed negligible acceleration which did not propagate inland.

### 3.3.3 Category 3: Sustained periods of glacier advance

Four glaciers underwent sustained periods of glacier advance and showed overall deceleration from 1996 to 2016 (Table 1).
In contrast to Category 1 and 2 glaciers, periods of advance coincided with acceleration (Figure 9). This behaviour was most distinct at Ryder Glacier, which accelerated by ~8% (4.7–5.5 m a$^{-1}$) during both 7-year periods of glacier advance (Figure 9h). Between these periods of advance (~2 years), large retreat events took place (e.g. 2006–2008: Figure 6c), which coincided with some deceleration (-27 m a$^{-1}$). While velocity data are absent for the periods of early advance at Storstrømmen and L. Bistrup in the 1970s, we note similar velocity behaviour during the more recent record (Figure 9). In contrast to most other outlet glaciers in northern Greenland, velocities at Storstrømmen and L. Bistrup Bræ are fastest inland, and decrease towards the terminus (Figure 9b,c). However, throughout the record (1996 to 2016), grounding line terminus velocities accelerated by 350% and 150% at Storstrømmen and L. Bistrup Bræ (Figure 9); and velocities ~20–40 km inland decelerated by 10–15 m a$^{-1}$ (> 54%). A similar pattern of terminus acceleration and inland deceleration took place at Kofoed-Hansen Bræ, which drains the northern branch of Storstrømmen (Figure 7l).

### 3.4 Surface elevation change

### 3.4.1 Category 1: Minimal change, followed by steady retreat

Thinning rates on all Category 1 outlet glaciers with grounded-termini (except Kofoed-Hansen Bræ) increased between the period 1992–1996 and 2014–2015 (Table 1). Short-term (1–2 years) surface lowering was synchronous with the start of their steady retreat and clear examples of this were at Marie-Sophie and Academy Glaciers. A reduction in thickening rates (~0.06 m a$^{-1}$: 1999 to 2000), was followed by high retreat rates in the following years at both Marie-Sophie (-130 m a$^{-1}$: 2001 to 2004) and Academy Glacier (-205 m a$^{-1}$: 2001 to 2003). Periods of greater retreat (2001 to 2003/04) were then followed by dramatically increased thinning rates at both glaciers to -0.16 m a$^{-1}$ (Marie-Sophie) and -0.3 m a$^{-1}$ (Academy: Figure 7i,k). Thinning rates similarly increased strongly from -0.19 m a$^{-1}$ to -0.78 m a$^{-1}$ at Humboldt Glacier from 1996–2005 to 2005–2012, which coincided with increased retreat rates (-98 to -160 m a$^{-1}$). Limited SEC data prevent us from commenting in depth on elevation changes at glaciers in NW Greenland. However, the few years of data available at Harald Moltke Bræ show increased thinning between 2012 and 2015, coincident with retreat (Figure 7c, g). Within this record lies an anomalous year of reduced thinning rates (2013 to 2014), which were coincident with an order of magnitude increase in velocity (~1000 m a$^{-1}$) and 0.8 km terminus advance. A single exception to increased thinning rates at grounded-terminus glaciers was Kofoed-Hansen Bræ, where thinning extends only ~20 km inland of the terminus, before it switches to thickening further inland (Figure 7l).



### 3.4.2 Category 2: Minimal change, followed by rapid, short-lived retreat

In comparison to grounded-terminus glaciers in Category 1, those with floating ice tongues in Category 2 experienced even higher thinning rates from 1992–1996 to 2014–2015 (Table 1), and were characterised by short-lived dramatic increases in thinning rates following ice tongue retreat. This was clear at Petermann, Hagen Bræ, and Zachariae Isstrøm, which all showed a slight thickening before ice tongue retreat/collapse, followed by a clear switch to thinning immediately before large calving events. For example, rates of elevation change at Petermann Glacier switched from negligible thickening in 2008 (0.03 m a$^{-1}$) to thinning (-0.22 m a$^{-1}$) in 2009, before the removal of 27 km of floating ice in the following three years (2010 to 2013). At Zachariae Isstrøm a clear switch to thinning was also synchronous with the onset of rapid retreat in 2003 (Figure 8h, k), although thinning rates increased more dramatically once the entire ice tongue was lost between 2011 and 2012. Thinning rates during and immediately after floating ice tongue retreat increased from minimal change (< -0.2 m a$^{-1}$ thinning) to -0.8 m a$^{-1}$ at Petermann Glacier (2010 to 2013), -1.7 m a$^{-1}$ at Hagen Bræ (2007/11 to 2011/12), and -4.3 m a$^{-1}$ at Zachariae Isstrøm (2011 to 2012). In all cases, dramatic increases in thinning rates were also coincident with acceleration during the years following ice tongue removal (Figure 8). Other glaciers showed more gradual and less dramatic increases in thinning rates (Figure 8). For example, at C. H. Ostenfeld the removal of 21 km of floating ice between 2002 and 2003 was followed by a steady and low magnitude increased thinning rates at a rate of -0.04 m a$^{-1}$ from 2003 to 2011 (Figure 8i). In this case, velocity increases alongside increased thinning rates were also gradual, but minimal in comparison to other glaciers.

### 3.4.3 Category 3: Sustained periods of glacier advance

Surface elevation data coverage was more limited at these glaciers in northern Greenland, so we focus on those with the most complete record (Storstrømmen and L. Bistrup Bræ). Storstrømmen and L. Bistrup Bræ thinned at the glacier terminus and thickened inland from 1996 to 2015 (Figure 9b, c). This was also partly seen at Kofoed-Hansen Bræ (Figure 7l). Periods of glacier advance (~1970s–80s) at both Storstrømmen and L. Bistrup Bræ preceded the earliest record of SEC and, following this, their terminus positions underwent minimal change (Figure 6). Between 1996 and 2015, inland elevation change was minimal (Figure 10), whereas greater thinning took place at the terminus. Large retreat events of 2.1 km at Storstrømmen and 0.7 km at L. Bistrup Bræ between 2011 and 2013 coincided with increased terminus thinning rates of -0.8 m a$^{-1}$ at Storstrømmen (2011 to 2012) and -1.76 m a$^{-1}$ at L. Bistrup Bræ (2011 to 2013: Figure 10). The spatial pattern of elevation changes were synchronous with velocity variations: deceleration and thickening occurred inland, while acceleration, thinning, and retreat were synchronous at the terminus (Figure 9). Despite poor SEC data at Ryder Glacier (Figure 9d), it is possible to comment on elevation changes during periods of recent advance (1999 to 2015). Surface elevation change 34 km inland of the grounding line switched from thickening during 1996 to 2002 (0.04 m a$^{-1}$) to substantial thinning from 2003 to 2006 (-2.7 m a$^{-1}$). Increased thinning rates coincided with an ~8% increase in ice velocities which preceded a large 3.2 km calving event in 2006 (Figure 9h). Following this retreat event, elevation change inland switched back to thickening of 0.1 m a$^{-1}$ by 2007, which was coincident with deceleration of -27 m a$^{-1}$ from 2006 to 2008.



### 3.5 Climate-ocean forcing

Annual air temperatures from Pituffik (NW Greenland) and Danmarkshavn (NE Greenland) range between -7.9 and -13.6°C from 1948 to 2015, and showed a significant increasing trend (p-value < 0.01) from the early 1990s onwards (Figure 11). At both stations, changepoint analysis revealed a significant change in mean annual air temperatures around 2000 (Figure 11).

Annual air temperatures were 1.4°C warmer from 2000 to 2015 than mean temperatures of -11.1°C (Pituffik) and -12.1°C (Danmarkshavn) from 1948 to 1999. While annual air temperatures show this clear break, summer temperatures (JJA) remain relatively constant at both stations (Figure 11). Before the change-point in 2000, average annual positive degree days (PDDs) at Danmarkshavn were 254 (1958 to 1999) and 364 at Pituffik (1974 to 1999). After 2000, mean PDDs then increased to 352 at Danmarkshavn from 2000 to 2015 and increased even more dramatically to 667 at Pituffik from 2000 to the end of the PDD

record (2006: Figure 11). Thus, a clear difference between NE and NW Greenland exists, where air temperatures and PDDs are greater in the northwest (Pituffik: Figure 11).

Coincident with increased air temperatures from 1990s, decadal sea ice concentrations were anomalously low across northern Greenland from 1986 to 1995 relative to the overall mean from 1979 to 2015 (Figure 12b). At the NEGIS, in particular, SICs

remained up to 20% lower than the 1979 to 2015 mean throughout the following decade (1996 to 2005: Figure 12c). In the final decade (2006 to 2015), this was followed by a switch to greater SICs in the northeast region, and a reduction in sea ice concentrations in the NW in comparison to the 1979 to 2015 mean (Figure 12).

### 3.6 Topographic factors

Distinct variability in glacier geometry exists between outlet glaciers in northern Greenland. Glaciers with grounded termini

in Category 1 tend to be characterised by deep beds (-33 to -370 m below sea level). Several of these catchments rest on reverse bed slopes (e.g. Heilprin, Tracy and Harald Moltke Bræ: Table 2), but several others have relatively flat bed profiles (e.g. Marie Sophie, Humboldt, Academy: Figure 7i-j). Catchments that have the largest areas resting below sea level are Harald Moltke Bræ (17%) and Humboldt (27%: Table 2). Grounded-terminus glaciers in Category 1 are also mainly confined within long narrow fjords (5–16 km wide), which widen inland (Table 2). Most Category 2 glaciers with floating ice tongues have

deeper bed topography (-73 to -480 m below sea level) and all lie on retrograde bed slopes (Table 2). Several catchments also have large proportions that rest below sea level, particularly the NEGIS (54%) and Petermann Glacier (67%). While all Category 2 glaciers have overall inland sloping beds, their current grounding line positions vary between resting on steep sloping sections (e.g. Zachariae Isstrøm and Steensby), to relatively flat topography (e.g. Hagen Bræ and C. H. Ostenfeld: Figure 8). Fjord widths are, on average (21 km), wider than Category 1 glaciers, although some terminate in long narrow fjords

(Petermann, Steensby, C. H. Ostenfeld), while others are less confined by fjord walls (Nioghalvfjerdsfjorden and Zachariae Isstrøm). Glaciers which have shown periods of sustained advance (Category 3) also rest below sea level, reaching a maximum of ~1000 m near the grounding line at Ryder Glacier (Figure 9d). All four glaciers show basal depressions, which reach depths



~26% lower than the rest of their basal profiles (Figure 9). Fjord widths vary greatly for these glaciers, but on average are narrower than Category 2, but wider than Category 1 (17 km).

## 4. Discussion

### 4.1 Timing of glacier change and climate

Decadal retreat rates between 1948 and 2015 (Figure 4) show a clear transition to overall retreat in 1995: average front position change switched from +72 m a$^{-1}$ (advance: 1948 to 1995) to -445 m a$^{-1}$ (retreat: 1996 to 2015). This includes the onset of steady retreat at most grounded outlet glaciers in northern Greenland, and the occurrence of large, rapid retreat events at floating-ice tongue glaciers (Figure 6). Increased decadal rates of terminus change across northern Greenland from 1995 coincided with increased air temperatures from the 1990s–2000s onwards (Figure 11). After the year 2000, mean air temperatures were 1.4°C

warmer compared to the 1948–1999 average, in both the northwest and northeast regions of the GrIS (Figure 11). These changes coincide with Arctic-wide increased retreat rates (Carr et al., 2017b), acceleration and retreat in south-east Greenland (Howat et al., 2008; Seale et al., 2011), and, to some extent, recent changes in north-west Greenland (e.g. Carr et al., 2013; Moon et al., 2012).

Increased thinning rates have taken place in the ablation areas (< 2000 m elevation) around the GrIS since the 1990s (Abdalati et al., 2001; van den Broeke et al., 2016; Krabill et al., 2000). At several glaciers, e.g. Jakobshavn (Thomas et al., 2011), and Helheim and Kangerdlugssuaq in the south-east (Howat et al., 2008; Luckman et al., 2006), linearly increasing temperatures after the 1990s increased thinning in the ablation zone, which reduced basal/lateral drag and instigated terminus retreat. In northern Greenland, it is likely that similar increased ice marginal thinning due to negative mass balance (van den Broeke et

al., 2016; Khan et al., 2015; Pritchard et al., 2009), may have been the initial condition for increased glacier retreat rates and feedbacks between retreat, acceleration and further dynamic thinning. Another subsequent feedback mechanism may have been surface melt induced hydrofracture, either through water filled crevasses (Benn et al., 2007; Nick et al., 2010), or supraglacial lake drainages through the full ice thickness (e.g. Banwell et al., 2013; Carr et al., 2015). Examples of where this might have been the case are at Zachariae Isstrøm and C. H. Ostenfeld, where water filled crevasses are clearly present along

their ice tongues before collapse. Additionally, many ice tongue and grounded-terminus outlet glaciers supported supraglacial lakes throughout the summer months (e.g. Humboldt: Carr et al., 2015) and, particularly in the northeast, they are likely to become even more common in the future (Ignéczi et al., 2016).

Another key impact of warmer air temperatures from the 1990s onwards across northern Greenland is the removal of sea ice

from the fjords (Figure 12). Sea-ice buttressing has previously been identified as an important control on glacier calving rates, both at glaciers in northern Greenland (e.g. Higgins, 1990; Johannessen et al., 2013; Khan et al., 2014) and elsewhere (e.g. Miles et al., 2016; Moon et al., 2012, 2015). The NEGIS has been highlighted as a region particularly sensitive to sea ice



changes (Khan et al., 2014; Reeh et al., 2001). Sea ice concentrations in this region decreased from 1996 to 2005 (Figure 12c), and coincided with periods of retreat at both glaciers (Figure 4d). At Nioghalvfjerdsfjorden, in particular, it is likely sea ice removal allowed major calving to occur from 1995 (Reeh et al., 2001) until the end of our study. Elsewhere in the region, the removal of sea ice from 2005 to 2015 in the NW (Figure 12d), coincided with greater retreat from 2005 onwards at Humboldt

Glacier (Figure 7n). This supports previous assertions that many outlet glaciers are highly sensitive to changes in sea-ice buttressing (Amundson et al., 2010; Carr et al., 2015). Across the most northern regions of the study area, from Petermann Glacier east to Hagen Bræ, anomalously low sea ice conditions occurred between 1986 and 1995 (Figure 12b), which coincided with the onset of some retreat during this period (Figure 4c). Similar to early work (Higgins, 1990), this suggests sea ice removal from the fjords can allow the removal of calved ice away from the terminus and increase the length of the 'calving

season'.

An important area of future work in northern Greenland, is understanding the role of ocean conditions in controlling outlet glacier behaviour. Elsewhere increased ocean temperatures have coincided with acceleration, retreat and thinning (e.g. Moon and Joughin, 2008; Straneo and Heimbach, 2013). Despite not being exposed to warm subtropical waters like elsewhere around

Greenland (e.g. east Greenland: Seale et al., 2011), increased ocean temperatures can markedly increase basal melt rates on large floating tongues (Mouginot et al., 2015; Reeh et al., 1999; Rignot et al., 2001, 1997). Early work highlighted the importance of basal melting for the mass balance of ice-shelves (Reeh et al., 2001; Rignot et al., 2001; Rignot and Steffen, 2008) and melt rates beneath the three remaining floating ice tongues (Nioghalvfjerdsfjorden, Petermann, and Ryder) are estimated to exceed ~50 m a$^{-1}$, which is >80% of the total melt flux at all three glaciers (Wilson et al., 2017). While ocean

warming is another likely control on dynamic glacier change, we are unable to make a more objective assessment on this due to limited region wide ocean temperature data. With the availability of more spatially extensive ocean/fjord temperature data in future, more focus on the role of ocean warming on glacier change in northern Greenland is needed.

### 4.2 Dynamic glacier response to terminus change

Our analysis has revealed three broad categories of terminus change for the period 1948–2015 for 18 major outlet glaciers in

northern Greenland: (1) those that underwent minimal change followed by steady retreat; (2) those that underwent minimal change followed by rapid, short-lived retreat; and (3) those that underwent a period or several periods of sustained advance. Importantly, we find that Category 1 corresponds to those glaciers with grounded termini and Category 2 corresponds to those with floating ice tongues. Category 3 includes several glaciers that we interpret to be surge-type, and their potential surge-type behaviour are discussed separately in Section 4.4. Moreover, our results show that the dynamic response to a calving front

perturbation/change is highly dependent on whether the terminus is grounded or floating. Here we discuss these differences between glaciers with floating versus grounded termini.



Both grounded and floating-terminus glaciers showed increased thinning in the years prior to retreat. As such, thinning may have initiated accelerated terminus velocities, thinning, and enhanced rates of retreat, as in other regions of the ice sheet (e.g. Luckman et al., 2006; McFadden et al., 2011; Moon and Joughin, 2008). Following initial thinning at the terminus, grounded-terminus outlet glaciers in northern Greenland (Category 1) underwent prolonged periods of steady retreat (on average -150 m a$^{-1}$) that usually lasted for two to three decades (Figure 6b). During these steady retreats, annual ice velocities increased by 27–110%, and surface thinning rates increased (Figure 7). In several cases, there was a clear inland propagation of accelerated flow following retreat (e.g. Heilprin and Tracy Glaciers: Figure 7a, d). Steady and continuous retreat accompanied by prolonged acceleration and thinning is analogous to grounded-terminus outlet glaciers elsewhere e.g. Helheim and Kangerdlugssuaq (Howat et al., 2008, 2005, 2007) and in west Greenland (McFadden et al., 2011). Sustained terminus retreat likely caused a large and prolonged stress perturbation at the terminus, which allowed acceleration and thinning to propagate inland and continue for a longer period before the glacier reached a stable geometry (McFadden et al., 2011; Nick et al., 2009).

In contrast to periods of steady retreat, terminus changes at floating ice tongue glaciers (Category 2) were characterised by short-lived (<6 years), high-magnitude retreat events that averaged -4536 m a$^{-1}$ (Figure 6a), and the dynamic response was more variable. In most cases rapid, large calving events were followed by either minimal and/or short-lived increases in annual velocity, and short-term increases in ice surface thinning rates (Figure 8). This contrasts with the behaviour of ice-tongue terminating glaciers elsewhere in Greenland (e.g. Joughin et al., 2008) and glaciers draining into Antarctic ice shelves (e.g. Scambos et al., 2004), which instead showed prolonged acceleration and dynamic thinning following the loss of substantial floating ice. For example, little dynamic change was seen at C. H. Ostenfeld and Petermann Glaciers in response to entire ice tongue collapse or large calving events. At other glaciers (Steensby and Hagen Brae) short-lived acceleration (< 13%) occurred near the terminus following ice tongue collapse, before a rapid return to pre-retreat velocity. On these glaciers, the minimal/short-lived dynamic response suggests that the stress perturbation associated with losing substantial sections of the ice tongues was minimal. In contrast to episodic calving events at the majority of floating tongue glaciers, gradual and sustained annual calving at Zachariae Isstrøm (ZI) was accompanied by a longer period of glacier acceleration and thinning, similar to the response of grounded northern Greenland glaciers, and conforms to the behaviour of ice-tongue terminating glaciers elsewhere (e.g. Jakobshavn Isbræ: Joughin et al., 2004, 2008). With the exception of ZI, our data show outlet glaciers in northern Greenland have been largely insensitive to either entire ice tongue loss (C. H. Ostenfeld, Steensby and Hagen Brae), or large iceberg calving events (Petermann, Nioghalvfjerdsfjorden). The behaviour of glaciers with floating ice tongues contrasts strongly with grounded-terminus glaciers, which underwent a much larger dynamic response to small magnitude retreats (Table 1). This highlights the need to consider terminus type when assessing the long-term response of outlet glaciers to changes at their terminus.





### 4.3 Influence of glacier geometry

Variations in basal topography and fjord width have been previously identified as an important control on the dynamic response of glaciers in many regions of the GrIS (Carr et al., 2013; Howat and Eddy, 2011; McFadden et al., 2011; Thomas et al., 2009; Carr et al., 2017). In this study, the differences between periods of retreat, acceleration, and thinning between floating and

grounded-terminus glaciers suggests basal topography may control the time taken for glaciers to return to a point where retreat slows and velocities return to pre-retreat levels. In the case of grounded-terminus glaciers (Category 1), prolonged acceleration and thinning following retreat suggests a long period of re-adjustment took place and was not complete by the end of the study period in 2015. This is likely due to deep basal topography (> 200 m below sea level), and retrograde bed slopes (~15 km of their grounding zones) beneath most grounded-terminus glaciers (e.g. Tracy, Heilprin, and Harald Moltke Bræ: Figure 7). We

suggest grounded-terminus retreat into deeper water contributed to: (i) buoyancy driven feedbacks, as the ice thinned to flotation (van der Veen, 1996), (ii) the penetration of basal crevasses through the full ice thickness (van der Veen, 1998, 2007), and (iii) subsequent enhanced rates of calving and continued retreat (e.g. Joughin et al., 2008). However, there are exceptions: retreat rates on Tracy Glacier substantially exceed those on Heilprin, despite the latter having steeper (i.e. higher gradient) inland-sloping basal topography (Figure 6: Porter et al., 2014). In this case, the deeper bed topography at Tracy Glacier (Figure

7d) promotes the intrusion of warm water to the glacier front (Porter et al., 2014), and basal topographic pinning points at Heilprin Glacier may have provided greater lateral drag and inhibited accelerated retreat down its deep sloping bed.

In contrast to grounded-terminus glaciers, Category 2 glaciers with floating ice tongues experienced a comparatively short-lived dynamic response to changes at their terminus. Ice tongue buttressing can importantly influence grounding line retreat.

It would appear that several floating ice tongues in northern Greenland provide limited buttressing, whereby ice loss at the tongue does not result in grounding line retreat or impact inland ice discharge. The response of these glaciers is also dependent on their bed topography and, in most cases, their grounding lines currently rest on relatively flat sections of their basal topography. Such relatively flat basal topography at Hagen Bræ and C. H. Ostenfeld (Figure 8c,i) could have prevented unstable grounding line retreat, and associated acceleration and thinning, following the loss of their floating ice tongues. In a

similar way, the flat sections of basal topography under Petermann Glacier and Nioghalvfjerdsfjorden could control their future response to ice tongue collapse (Figure 8b, g), as their grounding lines would need to retreat ~20 km inland in order to sit on a retrograde slope. In contrast, there has been prolonged thinning, acceleration and retreat following ice tongue loss at ZI, which can also be explained by basal topographic controls. Here, once the glacier became grounded, unstable retreat down a large basal over-deepening that extends ~20 km inland of the grounding line (Figure 8h) could have caused the positive

dynamic feedback response of acceleration and thinning (Khan et al., 2014; Mouginot et al., 2015). Overall, the different dynamic responses of floating and grounded-terminus glaciers to perturbations at their terminus, and their distinct basal topographic characteristics, highlights bed topography as a key control on the behaviour of glaciers in northern Greenland.



Fjord width and pinning points have both been identified as key controls on glacier response to forcing (e.g. Carr et al., 2013; Enderlin et al., 2013; Howat and Eddy, 2011; Jamieson et al., 2012), and could also explain differences between grounded-terminus and floating ice-tongue glaciers (McFadden et al., 2011), as well as individual glacier variability. At Hagen Bræ, for example, the ice tongue is confined by, and strongly attached to, its fjord walls, and retreat away from an island pinning point

may have reduced back stress on inland grounded ice, and contributed to its acceleration (Joughin et al., 2010). More limited dynamic responses at C. H. Ostenfeld and Nioghalvfjerdsfjorden may be due to the unconfined nature of the tongue within the fjord, and ice islands holding the tongue in place at Nioghalvfjerdsfjorden.

Overall, the minimal/short-lived dynamic response of most glaciers with floating ice tongues (Figure 8) suggests there was

limited buttressing forces acting on their terminus, and little resistance provided by the fjord walls. Instead, the calving of grounded ice which is strongly attached to the bed and fjord walls, is likely to have caused a larger stress perturbations due to the greater reduction of basal/lateral stresses at grounded-terminus glaciers (McFadden et al., 2011). The transfer of stresses is propagated inland, and drives accelerated ice flow and surface thinning, which may account for their more pronounced dynamic response to terminus retreat.

**4.4 Glacier surging**

In contrast to the majority of glaciers in northern Greenland, we identify four outlet glaciers which underwent sustained advance (Category 3) and suggest these are likely to represent surge-type glaciers. This is based on the following characteristics: 1) substantial periods of glacier advance (> 90 m a$^{-1}$) followed by retreat during the study period, 2) accelerated ice flow coincident with periods of advance, and 3) surface thickening inland and thinning at the terminus position indicative

of a quiescent surge-phase.

Ryder Glacier has the most complete record of a potential surge-type glacier in northern Greenland, and showed two clear periods of advance over the last two decades (1995 to 2015), and two further advances before 1995 (Figure 6, Figure 9h). Recent periods of advance were accompanied acceleration, and some surface thinning ~30 km inland of the grounding line, which provide strong evidence for it being surge-type. While our elevation data are limited, previous studies identified near-

terminus thinning (2–4 m a$^{-1}$: 1997 to 1999) and, at ~50 km inland, a similar magnitude of thickening (Abdalati et al., 2001), which is indicative of quiescent phase of surge-type glacier (e.g. Kamb et al., 1985; Meier and Post, 1969; Sharp, 1988). Despite the cyclic terminus behaviour and coincident changes in velocity that are indicative of a surge-type glacier, the long active phase (~ 7 years), followed by a short quiescence (~2–3 years) is in stark contrast to previously identified surge-cycle

timescales, during which the quiescent is usually far longer than the surge (e.g. Dowdeswell et al., 1991; Sevestre and Benn, 2015). Additionally, velocity increases were not dramatic during the surge (~8%). Thus, whilst this glacier is regarded to be surge type (Joughin et al., 1999; Rignot et al., 2001) and our evidence supports that, the short quiescence and small velocity increase are not typical of surge-type glaciers.





Alongside Ryder Glacier, we find further support for two other surge-type glaciers in northeast Greenland. Terminus changes recorded at Storstrømmen and L. Bistrup Bræ (Figures 6 and 9) confirm previous work that identified a surge event at Storstrømmen in the 1970s (Reeh et al., 1999). Interestingly both glaciers began to advance at a similar time, despite separate

drainage catchments, and advance continued until 1985 at Storstrømmen, and 1998 at L. Bistrup Bræ (Figure 6c). Unfortunately, velocity and surface elevation change datasets do not cover this period. However, dynamic changes at both glaciers between 1992 and 2016 were indicative of periods of quiescence. Both glaciers clearly show inland thickening, which coincides with slower glacier flow, and a terminus region of greater thinning, coincident with acceleration and retreat (Figure 9, 10). This confirms previous work that these glaciers are indeed surge-type (Abdalati et al., 2001; Csatho et al., 2014; Thomas

et al., 2009).

In northwest Greenland, Harald Moltke Bræ has been previously considered surge-type (Moon et al., 2012; Rignot and Kanagaratnam, 2006), and we record an additional surge event from 2013 to 2014, based on high magnitude acceleration ($\sim$1000 m a$^{-1}$) and glacier advance (0.8 km). This glacier fits the conventional definition of surging, i.e. a short active phase,

which included a clear order of magnitude increase in velocity (e.g. Meier and Post, 1969). However, it has a short surge-cycle (< 10 years) compared to most other glaciers in the Arctic (Carr et al., 2017a; Dowdeswell et al., 1991; Kamb et al., 1985), and underwent overall retreat from the late 1980s to 2015 (Figure 6), suggesting that climate-ocean forcing may be overriding its cyclical behaviour. Academy and Hagen Bræ glaciers have been previously identified as potentially surge type (Rignot and Kanagaratnam, 2006; Thomas et al., 2009), but we find no evidence to support this between 1948 and 2015. Brikkerne Glacier

is the final glacier within Category 3 that advanced between 1969 and 1978. However, our limited data on this glacier makes its surge behaviour less clear and the glacier could have instead been controlled by external forcing, or had a much longer surge cycle than can be seen from this record.

## 5. Conclusions

Outlet glaciers in northern Greenland drain $\sim$40% of the ice sheet by area but remain understudied compared to other regions

of the ice sheet. Here, we have analysed the dynamics of 18 major marine-terminating outlet glaciers in northern Greenland between 1948 and 2015. Overall, glacier retreat rates ranged from -15 to -311 m a$^{-1}$ over the entire study period. Between 1948 and 1995 glaciers exhibited generally low magnitude advance and retreat, with an average frontal position change of +72 m a$^{-1}$ (advance) across the 18 study glaciers. Following this, there was a clear regional transition to more rapid and widespread retreat, when average frontal position change was -445 m a$^{-1}$ (1995 to 2015). This was coincident with accelerated retreat in

other regions of the ice sheet (e.g. Carr et al., 2013; Howat et al., 2008; Moon et al., 2012). From 1996 to 2015, most glaciers also experienced accelerated ice flow and increased dynamic thinning. This switch in the mid-1990s corresponds to a regional increase in annual air temperatures from the 1990s onwards, and decreased sea ice concentrations. Thus, recent climate




warming may have been the precursor for initial thinning at the terminus of most outlet glaciers in the region, which initially forced subsequent acceleration, retreat, and thinning. Ocean temperature data is not readily available for this region, but ocean warming may also have played a role, as has been observed elsewhere in the ice sheet.

While increased retreat rates from the mid-1990s were near-ubiquitous, we observe distinct differences in glacier behaviour and these can be objectively categorised using statistical analysis. The first category are grounded-termini that experienced minimal change and then experienced a prolonged period of steady retreat. This likely resulted from a greater stress perturbation at the terminus that led to substantial acceleration and thinning. In contrast, the second category includes glaciers with floating ice tongues that were characterised by short-lived, high magnitude retreat events, which resulted in limited

dynamic glacier response. A key conclusion is that the dynamic response of outlet glaciers to perturbations at their calving front appears highly dependent on their terminus type. Glacier geometry (e.g. fjord width and basal topography) is also shown to be an important influence on the dynamic responses to terminus change. Continuous retreat at grounded terminus glaciers occurs down deep retrograde bed-slopes, and the removal of lateral resistive stresses from the retreat of grounded ice confined within narrow fjords could be responsible for prolonged acceleration and thinning. This also suggests a greater period was

taken for the glacier to re-adjust towards relative stability, where retreat and acceleration are reduced. The muted dynamic response of glaciers with floating ice tongues, instead suggests a more rapid yet short-lived readjustment took place. Glacier surging has not been systematically examined across northern Greenland, and we identify several surge-type glaciers in the region (Storstrømmen, L. Bistrup Bræ, Ryder, and Harald Moltke Bræ) that fall into our third category of glacier behaviour. In contrast to grounded or floating-terminus glaciers, surge-type glaciers were characterised by periods of substantial glacier

advance coincident with acceleration and surface thickening. Several glaciers showed short surge cycles (< 10 years) while others in the northeast of the region (Storstrømmen and L. Bistrup Bræ) are currently in quiescence and the length of their surge cycles is unclear. Given the recent loss of several ice tongues, and the recent acceleration in the retreat of numerous grounded marine-terminating glaciers we suggest that this region is undergoing rapid change, and could soon contribute an important component to sea level rise.



**Data availability**

Shapefiles of frontal positions for all 18 outlet glaciers in this study between 1948 and 2015 are freely available on request to the corresponding author. All other data sources, including: satellite imagery, historical maps, surface elevation change, annual velocity, climate and ocean, and topographic data, are already available online. The sources of each of these datasets are given

in the text and the supplementary information.

**Author contribution**

The initial project was designed by all authors, and E. A. Hill led the data analysis and interpretation, with comments throughout from all authors. E. A. Hill led the manuscript writing, and all authors contributed towards the editing of the manuscript and figures.

**Competing interests**

The authors declare that they have no conflict of interest.

**Acknowledgements**

This work was supported by a Doctoral Studentship award to E. A. Hill at Newcastle University, UK, from the IAPETUS Natural Environment Research Council Doctoral Training Partnership (grant number: NE/L002590/1). We thank the ESA for

granting access to SPOT–1 data (project ID: 32435). We are also grateful for Corona imagery which was digitised on request to the US Geological Survey. We acknowledge several free datasets used in this work. Historical map charts were made available from the Polar Geospatial Centre and the Perry-Castañeda Library at the University of Texas, Austin. We are grateful for Cryosat and ESA surface elevation change data, and Sentinel annual Greenland wide velocity maps made freely available as part of the ESA Greenland Climate Change Initiative. MEaSUREs velocity maps (Joughin et al., 2010), IceBridge

BedMachine v2 (Morlighem et al., 2014), and sea ice concentrations (Cavalieri et al., 1996) are available from the National Snow and Ice Data Centre (NSIDC). Landsat imagery were acquired from the US Geological Survey. Surface air temperature data were acquired from the Danish Meteorological Institute (Vinther et al., 2006).



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





**Table 1:** Summary data for 18 northern Greenland outlet glaciers, ordered according to the glacier category. These correspond to glaciers with grounded-termini, floating-termini, and potentially surge-type glaciers. Terminus change is calculated by subtracting 1948 and 2015 positions and converted to an annual rate. Velocity change is calculated by subtracting winter 1995/96 velocities from the most recent 2015/2016 along each glacier centreline profile and calculating the average. Similarly, surface elevation change (SEC) rates are differenced from the earliest 5-year running mean (1992 to 1996) and the most recent annual change of 2014 to 2015 and averaged along the glacier centreline.

| | Northern Greenland Outlet Glaciers | Terminus Change (1948–2015) (m a⁻¹) | Velocity Change (1995/96– 2015/16) (m a⁻¹) | Difference in SEC rates (1992– 1996 and 2014-2015) (m a⁻¹) |
|---|---|---|---|---|
| **Category 1: Grounded terminus** | Harald Moltke Bræ | -156 | 22.6 | |
| | Heilprin | -45 | 7.16 | -0.15 |
| | Tracy | -173 | 36.8 | -0.11 |
| | Humboldt | -111 | 0.32 | -0.51 |
| | Harder | -25 | 0.58 | -0.89 |
| | Marie Sophie | -15 | 1.03 | -0.43 |
| | Academy | -31 | -4.87 | -0.97 |
| | Kofoed-Hansen Bræ | -169 | -0.06 | 0.12 |
| **Category 2: Floating ice tongue** | Hagen Bræ | -162 | 6.45 | -0.83 |
| | Petermann | -311 | 3.78 | -1.34 |
| | Steensby | 2 | 2.59 | -0.33 |
| | C. H. Ostenfeld | -58 | 2.96 | -1.26 |
| | Zachariae Isstrøm | -282 | 20.3 | -2.98 |
| | Nioghalvfjerdsfjorden | -28 | 1.62 | -1.99 |
| **Category 3: Potentially Surge-type** | Ryder | 43 | -0.08 | 0.47 |
| | Storstrømmen | 41 | -1.11 | -0.18 |
| | L. Bistrup Bræ | 39 | -3.89 | 0.57 |
| | Brikkerne | 82 | -2.56 | |





**Table 2:** Glacier-specific factors at 18 northern Greenland study glaciers. Drainage catchments were derived from hydrological analysis of surface topography, and the percentage of the drainage catchment below sea level was calculated from bed topography data. Linear regression of bed topography data along glacier centrelines shows the bed slope direction beneath each glacier. Fjord width was measured at 500 m intervals and again linear regression established if each glacier fjord is widening/narrowing inland. Red and blue shading for bed-slope and fjord width represent expected instability, and stability respectively for each parameter.

| | Northern Greenland Outlet Glaciers | Drainage Basin Size (km²) | % of drainage basin below sea level | Inland bed-slope | Seaward bed-slope | Widening Fjord | Narrowing Fjord |
|---|---|---|---|---|---|---|---|
| **Category 1: Grounded terminus** | Harald Moltke Bræ | 666 | 17 | | X | | X |
| | Heilprin | 6,593 | 2.9 | X | | X | |
| | Tracy | 3,176 | 3.6 | | X | | X |
| | Humboldt | 51,815 | 27 | X | | *Does not terminate in fjord* | |
| | Harder | 792 | 0.2 | X | | | X |
| | Marie Sophie | 2,567 | 6.8 | X | | X | |
| | Academy | a | a | | X | X | |
| | Kofoed-Hansen Bræ | b | b | X | | X | |
| **Category 2: Floating ice tongue** | Hagen Bræ | 30,250[a] | 20 | X | | X | |
| | Petermann | 60,093 | 67 | X | | X | |
| | Steensby | 3,356 | 4.2 | X | | | X |
| | C. H. Ostenfeld | 11,013 | 1.5 | X | | | X |
| | Zachariae Isstrøm | 257,542[b] | 54 | X | | X | |
| | Nioghalvfjerdsfjorden | | | X | | | X |
| **Category 3: Potentially Surge-type** | Ryder | 36,384 | 40 | | X | X | |
| | Storstrømmen | b | b | X | | X | |
| | L. Bistrup Bræ | 26,660 | 4.4 | X | | X | |
| | Brikkerne | 929 | 2.3 | | X | | X |



# Figures

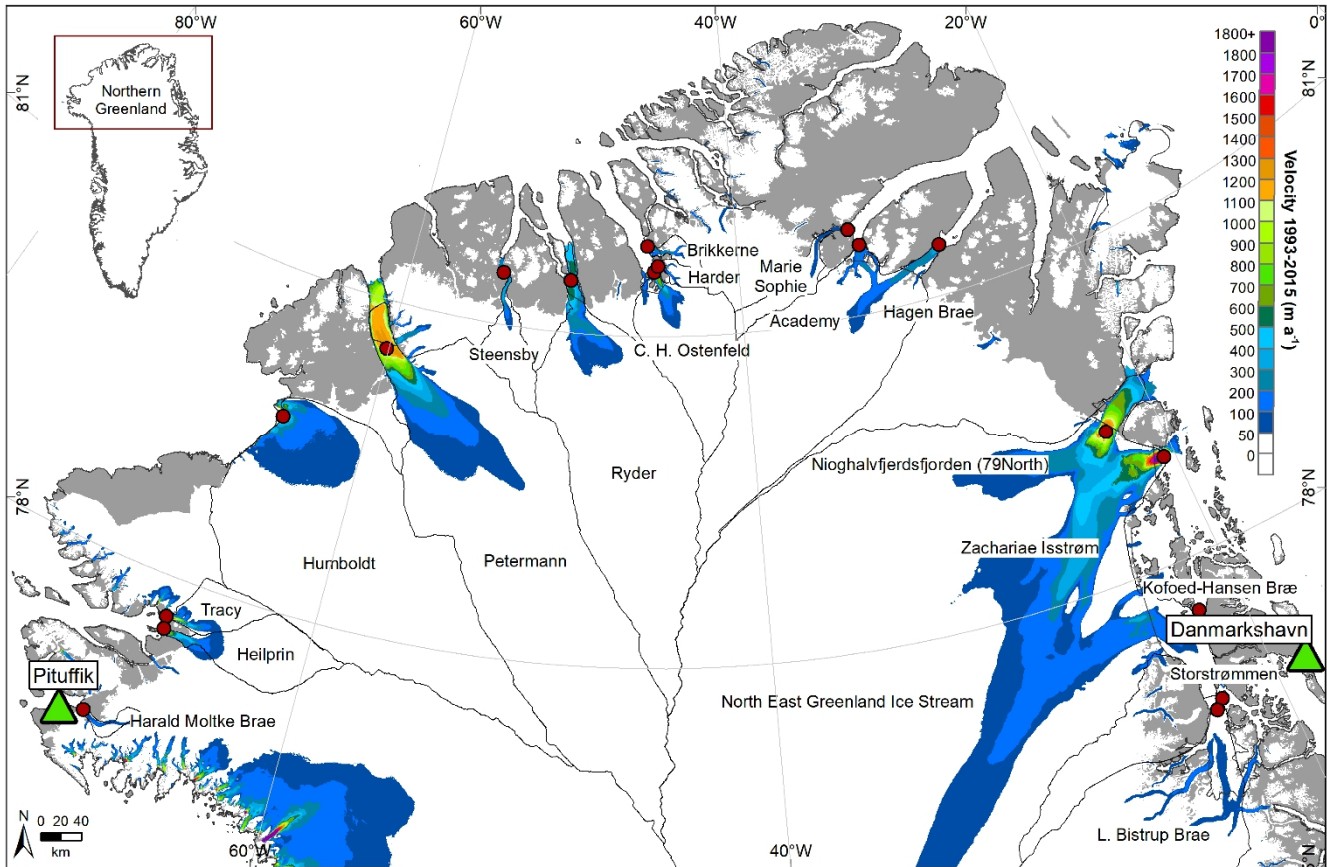

**Figure 1:** Study region of northern Greenland. Red circles show the location of each of 18 northern Greenland study outlet glaciers. Green triangles represent the location of Danish Metrological Institute automatic weather stations. Average glacier
5  velocities (m a$^{-1}$) are shown between 1993 and 2015 derived from the multi-year mosaic dataset (Joughin et al., 2010), acquired from the National Snow and Ice Data Centre. Glacier drainage catchments (black outlines) were calculated using the surface DEM from the IceBridge Bed Machine dataset (Morlighem et al., 2015), and hydrological analysis using the TopoToolbox in Matlab (Schwanghart and Kuhn, 2010).



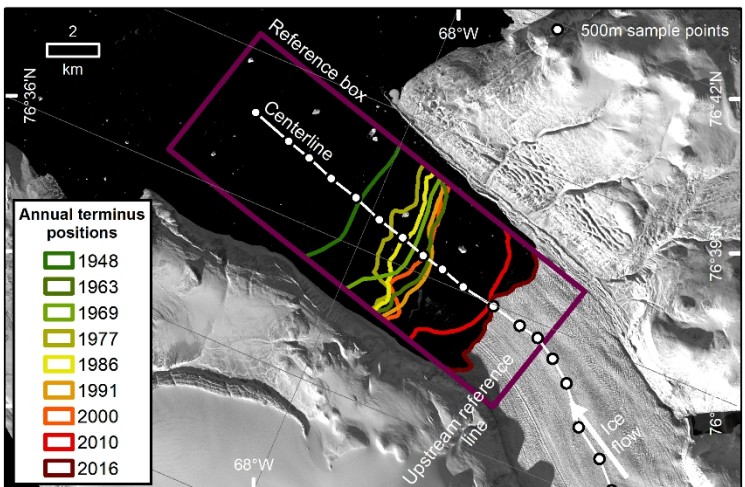

**Figure 2:** Rectilinear box method used to measure glacier terminus positions. An example at Harald Moltke Bræ, NW Greenland. The reference box is aligned parallel to the fjord sides, and joined inland of the furthest terminus position. Annual terminus positions (green to red) are digitised and joined at the upstream reference line. Glacier centreline profiles (white) are drawn down the centre of the glacier using Euclidean distance and sampled at 500 m intervals (white circles). Background image is Landsat 8 2016 panchromatic band 8 acquired from the USGS Earth Explorer.





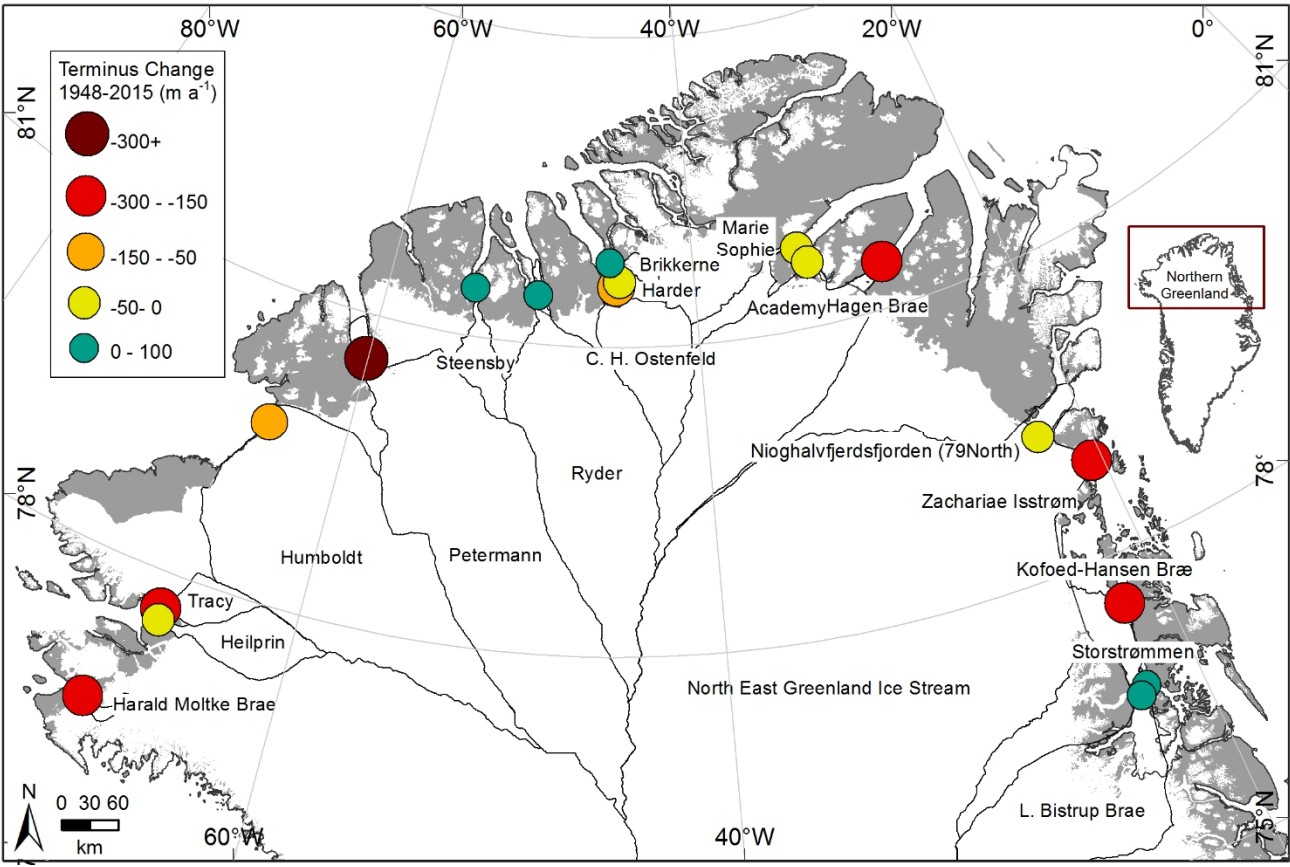

**Figure 3:** Overall rate of terminus change (m a$^{-1}$) at 18 outlet glaciers in northern Greenland from 1948 to 2015. Green circles represent glaciers which have undergone overall advance during the record, while yellow to red circles represent increasing retreat rates from 0 to larger than -300 m a$^{-1}$.







**Figure 4:** Mean decadal rates of terminus change (m a$^{-1}$) during five epochs between 1948 and 2015. Red circles represent glacier retreat rates between 0 and exceeding -1000 m a$^{-1}$ where the deeper red, and larger the circle the greater retreat rate. Blue circles represent advance rates between 0 and exceeding 1000 m a$^{-1}$ where darker blue, larger circles represent the highest rates of advance.





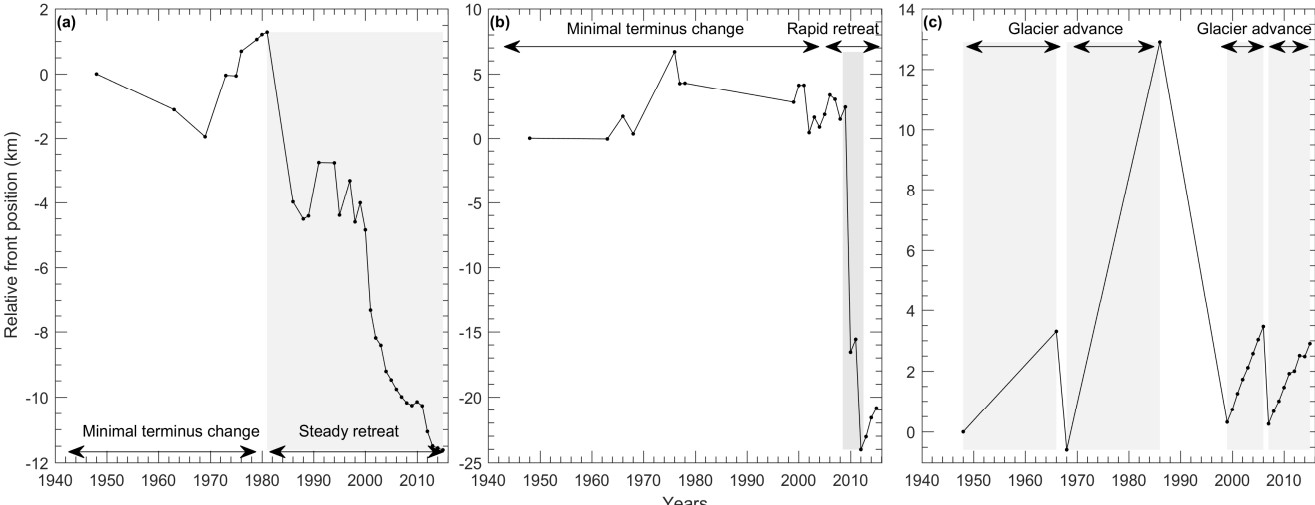

**Figure 5:** Schematic diagram of terminus position behaviour in northern Greenland. Three categories of behaviour determined from changepoint detection. The first (a) is an example of a glacier front position switching from minimal terminus change to a period of steady terminus retreat (grey), the second (b) shows a glacier which undergoes a period of rapid retreat (grey) after minimal terminus change, and the final (c) shows a glacier terminus position undergoing several periods of advance (grey), separated by periods of retreat.







**Figure 6:** Retreat rates during identified changepoint time periods. Glaciers are ordered by their retreat rates during their respective periods of rapid retreat, steady retreat and glacier advance A) Floating ice tongue terminating outlet glaciers, grey bars show their periods of minimal/variable terminus change (in some cases advance), and purple bars show the period of order of magnitude greater rapid retreat. B) Grounded-terminus glaciers, minimal/variable terminus change in grey and orange shows the period of steady glacier retreat. C) Potential surge-type glaciers, minimal/variable terminus change in grey and turquoise shows periods of substantial glacier advance.



**Figure 7:** Category 1 Glaciers: those which have undergone a switch from minimal terminus change to a period of steady retreat (Table 1). Top plots show annual surface elevation change (SEC) (m a⁻¹) during 1992–2015 (Green to Blue) with time. Middle plots show annual velocity change between 1991 and 2016 (yellow-orange-red). Below are surface, bed and floating ice tongue elevation profiles. Current (2016) terminus positions along these profiles are shown with a blue line. Surface and bed topography data were derived from the IceBridge Bed Machine dataset (Morlighem et al., 2014). Terminus position change at each glacier is shown on the bottom plots (black line).



**Figure 8:** Category 2 Glaciers; those which have undergone periods of rapid retreat (Table 1). Top plots show annual surface elevation change (SEC) (m a$^{-1}$) during 1992–2015 (Green to Blue) with time. Middle plots show annual velocity change between 1991 and 2016 (yellow-orange-red). Below are surface, bed and floating ice tongue elevation profiles. Current (2016) terminus positions along these profiles are shown with a blue line. Surface and bed topography data were derived from the IceBridge Bed Machine dataset (Morlighem et al., 2014). Terminus position change at each glacier is shown on the bottom plots (black line).





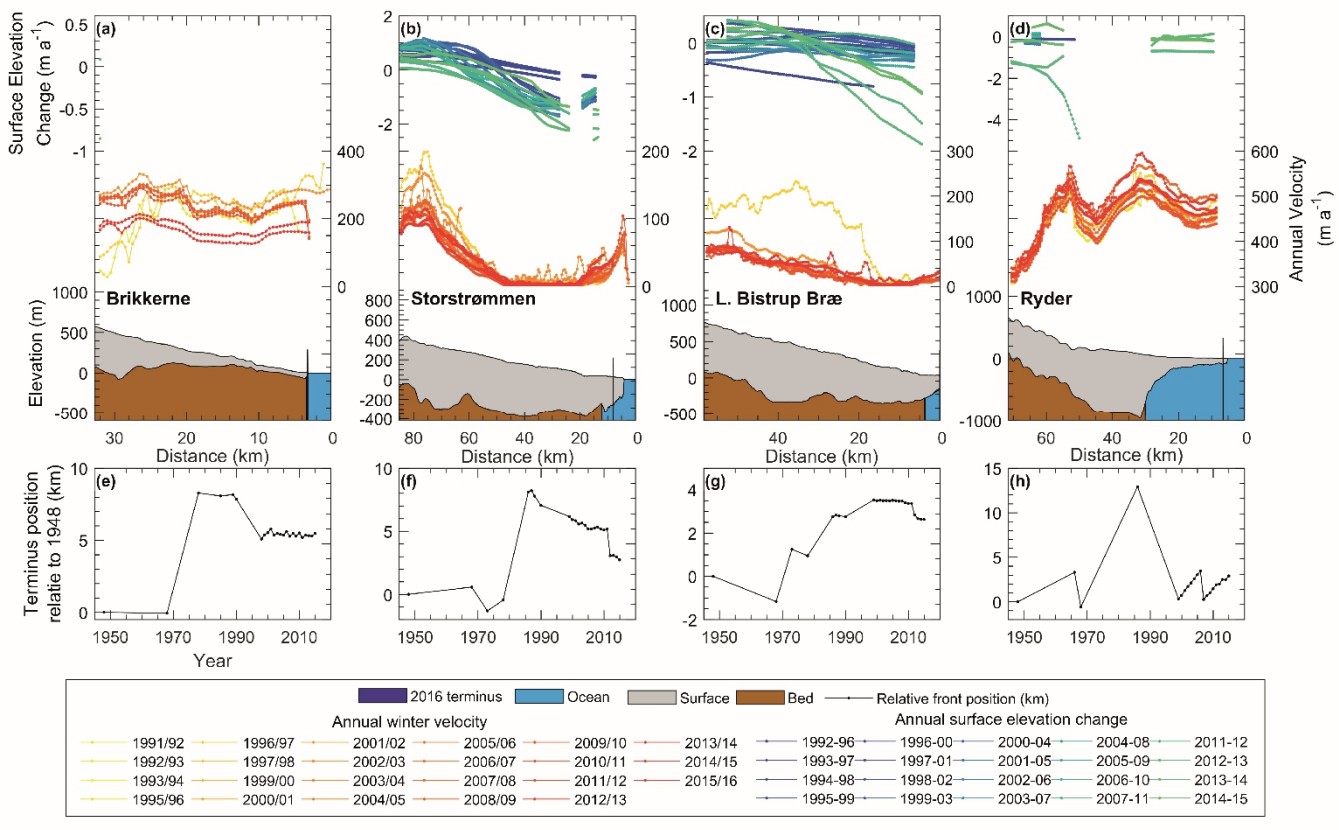

**Figure 9:** Category 3 Glaciers: those which have undergone periods of glacier advance (Table 1). Top plots show annual surface elevation change (SEC) (m a⁻¹) during 1992–2015 (Green to Blue) with time. Middle plots show annual velocity change between 1991 and 2016 (yellow-orange-red). Below are surface, bed and floating ice tongue elevation profiles. Current (2016) terminus positions along these profiles are shown with a blue line. Surface and bed topography data were derived from the IceBridge Bed Machine dataset (Morlighem et al., 2014). Terminus position change at each glacier is shown on the bottom plots (black line).



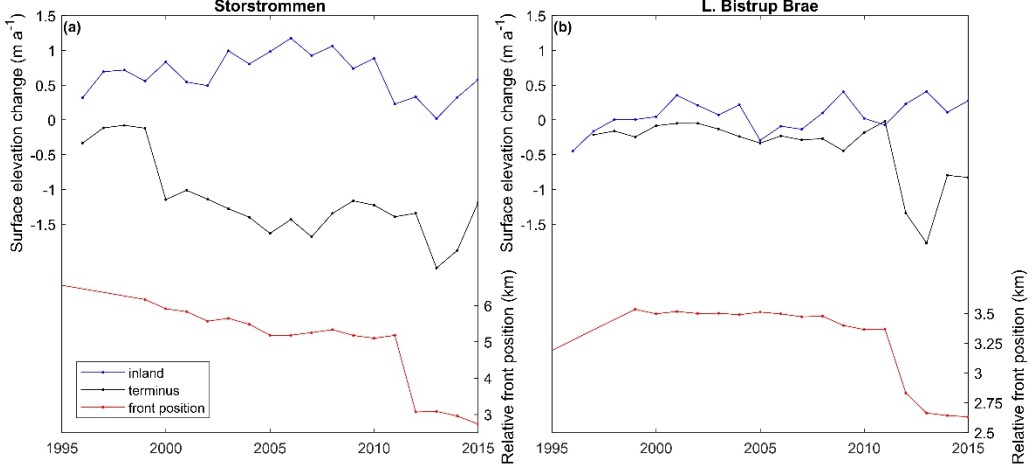

**Figure 10:** Surface elevation change at inland (blue) and terminus (black) locations between 1995 and 2015 for Storstrømmen (A) and L. Bistrup Bræ (B). Respective front position change during 1995 and 2015 are shown in red.

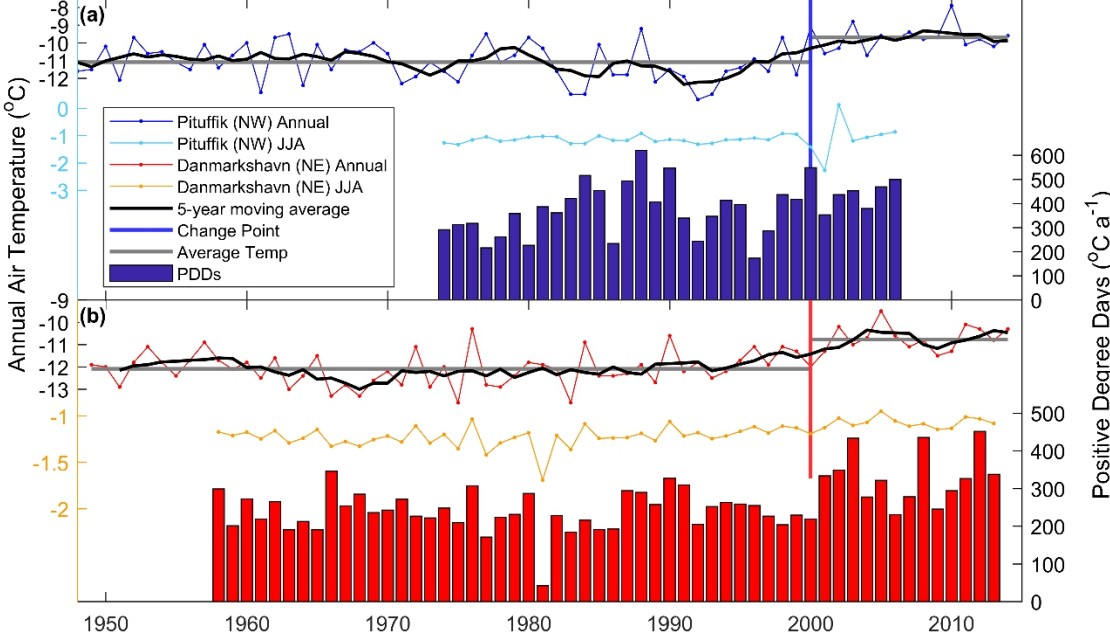

5  **Figure 11:** Climate-ocean forcing data for northern Greenland. A) shows annual air temperatures for Pituffik automatic weather station in NW Greenland (blue line) and calculated positive degree days from available daily air temperatures from the same station (blue bars). Summer temperatures (JJA) are also shown (light blue line). Change point break (thick blue vertical line) is accompanied with mean values for each epoch (grey). B) shows annual air temperatures for Danmarkshavn automatic weather station in NE Greenland (red line) and calculated positive degree days from available daily air temperatures

10  from the same station (red bars). Summer temperatures (JJA) are also shown (orange line). Change point break (thick red vertical line) is accompanied with mean values for each epoch (grey horizontal line). Both air temperature time series are fit with 5-year moving averages (black).





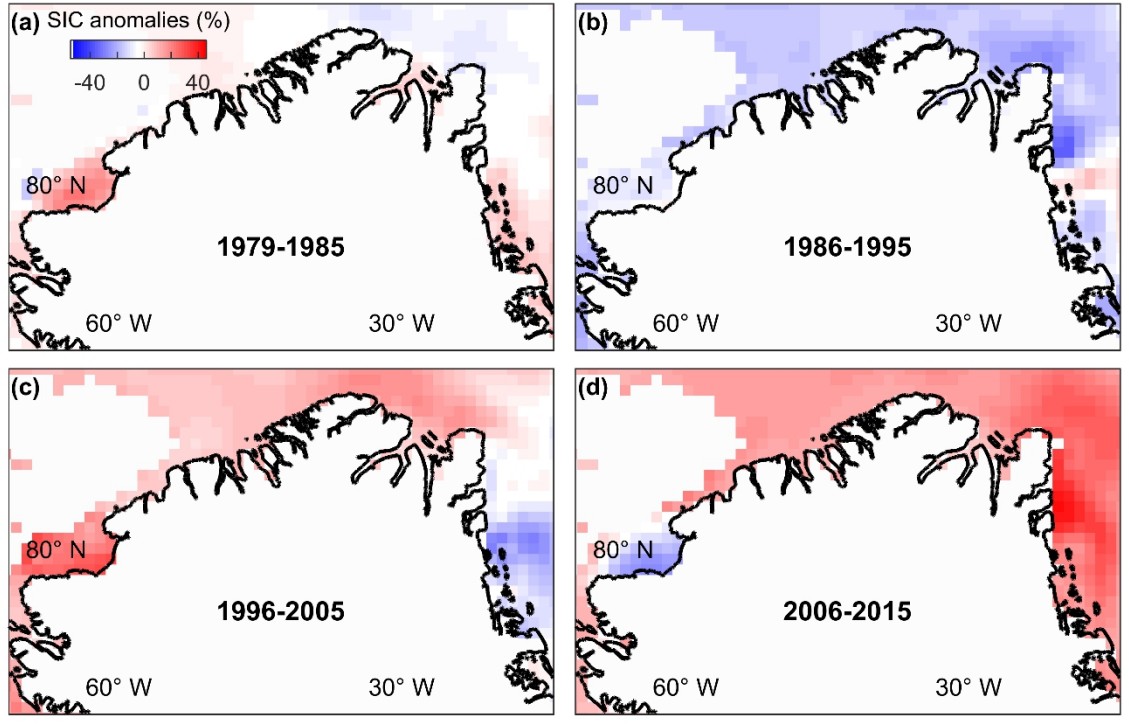

**Figure 12:** Sea ice concentration (SIC) anomalies (%) in relation to the 1979 to 2015 mean for four decadal periods across northern Greenland, except for the first to allow comparison with decadal terminus position changes. A) 1979 to 1985, B) 1986 to 1995, C) 1996 to 2005, D) 2006 to 2015.