# Peer review of "Dynamic changes in outlet glaciers in northern Greenland from 1948 to 2015"

_The Cryosphere, 2018_

## Referee Comment (RC1) · Anonymous Referee #1 · 20 Mar 2018

**Review of: Dynamic changes in outlet glaciers in northern Greenland from 1948 to 2015**

**General comments:**

This paper brings together data on glacier terminus position, speed, fjord geometry, and other metrics to examine glacier behavior across northern Greenland over 1948-2015. This is useful data to publish and results are in line with established ideas on glacier dynamics, influence of fjord geometry, and behavior of glaciers with or without floating ice tongues. Several tables figures are particularly useful for visualizing the results (e.g, Table 2 and Figure 6) and the paper adds new information about several glaciers and is quite thorough in addressing all marine-terminate northern glaciers.

Despite the strengths of the paper, there are fairly substantial areas for improvement:

In an attempt to pull climate and ocean conditions into the analysis, the authors include air temperature data from two weather stations and sea ice concentration from passive microwave (Section 2.4 and results in Section 3.5). The value of including these data seems extremely limited. On the air temperature side, only two weather stations are available, at the southern edges of the study area on the east and west coast. These data are used for a basic determination of changes in air temperature trend. For sea ice, the 25km resolution precludes analysis in narrow fjords or near the ice edge. It is well established that these data do poorly in capturing sea ice concentration at glacier termini in Greenland. Thus both the air and ocean data is severely lacking in detail compared to the other datasets the authors are working with. The authors even note themselves that they are focusing on ice dynamics and not air/ocean forcing (page 7, lines 3-5). I suggest that the authors reconsider the utility of these data and inclusion in the paper. They may instead choose to refer to data already published on Greenland air temperature and sea ice trends. The other analysis in the paper is of more interest and better quality.

At no point do the authors discuss some of the fundamental differences expected in glaciers with grounded termini versus floating ice tongues. I expected some acknowledgement that the former would have small, more continuous calving events and the latter would experience calving of large tabular icebergs. Since this is exactly what the authors observe, they need to provide some information and context for the behavior. This can also include a discussion of why smaller dynamic changes might be expected for glaciers with floating ice tongues. Without some of these notes, the results and discussion feel as though they have been pulled out of context from the greater body of glaciological literature.

The paper does suffer some overly complex sentences, wordy phrasing, and occasional poor organization. These items can be taken care of with mindful editing. Joshua Schimel's book Writing Science is an excellent reference for techniques and ideas.

**Specific comments by page/line number:**

1/12. 'remains unknown' is an overstatement and needs changing

1/23. This sentence is long and the wording at the end is overly complicated. Requires editing.

2/4. Moon et al. 2012 is a paper about ice speed and does not discuss thinning or retreat. This paper is incorrectly referenced in several places in the manuscript (e.g., also 14/32). An appropriate reference for thinning is: Csatho, B. M., A. F. Schenk, C. J. van der Veen, G. Babonis, K. Duncan, S. Rezvanbehbahani, M. R. van den Broeke, S. B. Simonsen, S. Nagarajan, and J. H. Van Angelen (2014), Laser altimetry reveals complex pattern of Greenland Ice Sheet dynamics, *Proceedings of the National Academy of Sciences*, *111*(52), 18478–18483, doi:10.1073/pnas.1411680112.

2/19-26. This paragraph would be better ordered: Sentence 2, sentence 1, sentence 3.

4/15-19. Another section that could be simplified/shortened. For example: 'Presence of sea ice and highly fractured termini made terminus picking at Steensby, CH Ostenfeld, and NGIS glaciers more difficult (Refs). Re-digitising all 1999-2015 Landsat terminus positions yielded additional errors of ±13% for these glaciers.'

5/3. It's not clear what range you are referring to – include the numbers here instead of 'this'.

5/3-5. This is confusing and I do not clearly understand the process from this description. Please revise.

5/11. It is better to refer to 'earlier' and 'later' instead of 'first' and 'second'.

5/12. Please specify what you are using to estimate average errors in velocity. This is more clear for other methods descriptions.

5/29-30. Why use only the difference between 1995/96 and 2015/16 velocity data to calculate change when you have so many years of data between these years. Seems that finding a trend across all years of data would provide a more accurate picture of change.

6/9-11. The same comment as above, but for the surface elevation change. Why use just two periods when you have more data in between? As a separate note, please reconsider using 'SEC'. This is not a commonly used acronym and the more you can avoid acronyms the easier it is to read.

6/30-31. It is not clear what using 'a flow accumulation threshold of 500 to calculate stream threshold' means. Please clarify.

7/30 and throughout manuscript. Remove 'clear'. This word is used widely throughout the paper and is superfluous. Recommend removing it in all cases.

8/8. Remove '1948-1975' from the first mention, and put these years in the second half of the sentence when you call out that the earliest epoch is 27 years long.

9/21-31. This description is poorly organized. I want a sense of what is happening at each glacier. Separate them out and talk about each with greater specificity. Describe how advance/retreat phases were more/less consistent and then changed (or not). How has the character of terminus change varied? I understand the urge to create something of a laundry list of information, and the difficulty into crafting fairly dry information into something that is easy to follow and structured across the paragraph. It is, however, important to work towards this goal. An good example of an organized, engaging description is page 12, lines 27-32.

10/22. 'Loss of their floating ice tongues' is incorrect for Petermann – instead just refer to 'retreat' or similar.

10/27. Something is not 'synchronous' with events in the following decade. Reword.

10/30-11/1. It's not clear if you mean changes in speed after large calving events or only after complete ice tongue removal. Please clarify.

12/2-15. 'Dramatic' appears several times in this paragraph – it's not a particularly useful or quantitative descriptor and I recommend revising/deleting. ('Clear' also appears several times in this paragraph).

13/19-14/2. Another paragraph in need of reorganization.

14/5 and 8. It is incorrect to refer to a single year (1995) as a change point because you are considering longer epochs. Refer to changes before/during/after those epochs rather than at specific years.

14/10. Clarify that 'These changes' is not referring to air temperatures.

14/15. This paragraph needs an introductory sentence and work on organization and flow.

14/26. The second half of this sentence is irrelevant to the discussion.

15/2-3 (and following paragraph). Acknowledge the role of other ocean processes, like ice front melt, in this sentence/section, followed by the more thorough discussion in the next paragraph. These references (or information within them) may be useful:
Wilson, N. J., and F. Straneo (2015), Water exchange between the continental shelf and the cavity beneath Nioghalvfjerdsbræ (79 North Glacier), *Geophys Res Lett*, *42*(18), 7648–7654, doi:10.1002/2015gl064944.
Choi, Y., M. Morlighem, E. Rignot, J. Mouginot, and M. Wood (2017), Modeling the Response of Nioghalvfjerdsfjorden and Zachariae Isstrøm Glaciers, Greenland, to Ocean Forcing Over the Next Century, *Geophys Res Lett*, *44*(21), 11,071–11,079, doi:10.1002/2017GL075174.

16/2. Write these in an order than makes more sense for the actual process, either thinning-retreat-speedup or retreat-speedup-thinning (use this latter one if you want the focus on dynamic thinning due to speedup).

16/15-19. It would be useful for the authors to comment on why they think these differences occur among the glaciers they mention. For example, how does scale of event and force balance based on glacier characteristics enter into the discussion. Also, it's not entirely clear whether the authors are consistently referring only to velocity changes on the grounded ice portion of these glaciers.

17/4. Another paper just out on this topic: Millan, R., E. Rignot, J. Mouginot, M. Wood, A. A. Bjork, and M. Morlighem (2018), Vulnerability of Southeast Greenland glaciers to warm Atlantic Water from Operation IceBridge and Ocean Melting Greenland data, *Geophys Res Lett*, 1–23, doi:10.1002/2017GL076561.

18/4 and 11. What do the authors mean by 'strongly attached to'? How has that been quantified, in this study or others?

19/18-19. A few more words are needed on this, and whether or not it is likely these are surge glaciers. Did you look at different data than these other studies? Can you definitely confirm there was no surge in periods where it was previously detected because you have better data or similar?

19/20. 'controlled by external forcing' is too vague. Say specifically what mechanisms might be at play and whether there is evidence for it, or what data would be needed.

19/30. Another incorrect reference to Moon et al. 2012. This would be a good place to reference Howat and Eddy 2010 (already listed in the references).

20/2. A variety of ocean data is available for northern Greenland. It is not, however, being used or analysed in this paper (which is just fine). But please remove this incorrect statement.

20/24. I understand the urge to end on 'could soon contribute an important component to sea level rise', but this is a vague statement and is not well connected to the paper analysis (which does not discuss sea level). Suggest rewording with a stronger concluding statement that is more specific and tied to the main idea of the paper.

27/3. This caption would benefit from more precise language throughout. The use of 'calculated by subtracting 1948 and 2015 positions' is one example.

Table 1. Consider the various order in which glaciers in each category could be listed and choose the one that makes the most sense for the reader or message.

Figure 1. The caption includes a lot of information on methods, which seems misplaced.

Figure 2. The legend should have lines rather than boxes.

Figure 4. Please reword for improved clarity and brevity.

Figures 7-9. It is very difficult to see the lines/colors in the legend and in the plots. Distinguishing among the surface elevation change lines to understand their progress is only possible in a broad green or blue sense. Understanding the detailed progress is impossible with the current color map.

Figure 8. Remove the odd floating ice in 8h, which does not appear to be connected to the glacier.

Figure 9. Is there no data for showing terminus position in 9c?

Figure 10. Instead of 'inland' and 'terminus' give a number for actual location/distance.

Figure 11. It's quite odd to stack the warmer temperatures below the colder temperatures in these plots. You also mention 'ocean' in the caption data, which is not included in the plots.

**Technical corrections by line number:**

2/5. Delete 'across the ice sheet' – unnecessary.

2/30. Delete 'objectively' – unnecessary.

7/30. Delete 'eventual'

10/8. Delete 'It was also clear that'. I'm not going to note anymore of the instances of 'clear', but just repeat that they should all be removed.

11/19. Thickening or thinning?

11/21. Delete 'then'

15/7. 'concentrations' instead of 'conditions'

15/9. Remove quotes around calving season.

15/11. Remove ','

17/19. 'importantly influence' is very awkward – reword

18/24. Should be 'accompanied by acceleration'

19/17. 'overriding' is poor word choice – please change

35/5. Replace 'Current' with '2016'

Figure 12. Delete 'except for the first…position changes'

---

## Referee Comment (RC2) · Anonymous Referee #2 · 27 Mar 2018

This manuscript presents a large volume of data for several glaciers in N. Greenland to draw conclusions about their dynamic behaviour over a significantly long time period. The data presented are of some value, but I'm afraid that this manuscript suffers a bit from explaining everything without really explaining anything. What I mean is that there is a dizzying array of information about climate, topography, glacier behavior to keep track of but not one factor comes across as being important to explaining the behavior of all glaciers. It's a challenge for the reader to keep track of all of the information and to make sense of what facts are important throughout the text.

The other issue I have is with the categorization of glaciers. There are categories 1) grounded terminus; 2) floating ice tongue and 3) potentially surge-type. Two of these reflect the state of the terminus while the last one reflects the inherent dynamics inferred from terminus behavior. Further, several of Category 3 have (or had) floating tongues, making it a challenge to keep up with the author's thoughts at times. Then, there is a second categorization – based on retreat style: 1) steady retreat; 2) rapid retreat; 3) advance. It's just too much to keep track of. In the conclusions, the authors say that "a key conclusion is that the dynamic response of outlet glaciers to perturbations....depends on their terminus type". However, with such poorly organized material and categorization it's unclear how this conclusion is supported.

In addition, the category of "sustained advance" is completely untrue. These glaciers (shown in 6c) undergo periods of advance AND retreat, in some cases, very rapid retreat, which is a far cry from sustained advance. This type of behavior is typical for glaciers with floating tongues – see MacGregor et al., JGlac 2012 58(209) and other tidewater glaciers – see McNabb et al., JGR-ES 2013 for additional examples of this. I feel as though categorizing these glaciers as "surge type" is a bit "getting off too lightly" – there is likely more to explain here. The authors would benefit from a close read of Steiger et al. (Cryosphere 2017) that suggests pinning points having an impact on glacier terminus positions. Also, the authors should examine the tidewater glacier cycle literature which discusses quite broadly the idea of cyclic glacier changes.

The authors use BedMachine v2, when BedMachine v3 has been released now for a year. v3 represents significant improvements, particularly in the terminus regions because of the addition of bathymetry data from the OMG project. It would be useful to know how the authors determined if the bed data were good or not. Some glaciers were not sufficiently sampled with radar data for the mass-conserving solution and thus, are not well-constrained in BedMachine. Finally, very little information is provided about how the authors calculated bed slopes at the glacier termini and how bed topography is used in general. The mention of pinning points and the comparison between slopes of the beds of glaciers is described with no data presented.

In the discussion, the authors invoke processes such as increased ablation rates, water drainage to the ice bed, and the removal of sea ice to explain the timing of glacier retreat. However, these correlations are presented as anecdotes, with very little in the way of evidence suggesting cause/effect. They discuss topography as well stating that bed topography is a "key control on the behaviour of glaciers in northern Greenland" but provide very little in the way of evidence for the reader to understand how this conclusion came to be. Bed topography is inherently three-dimensional and so presenting the topography in the small-scale images in the figures is not sufficient evidence for the reader.

Some additional edits are made in-line with the text in the attached pdf, but towards the middle I stopped correcting small things.

Please also note the supplement to this comment:
https://www.the-cryosphere-discuss.net/tc-2018-17/tc-2018-17-RC2-supplement.pdf

**Supplement:**

[revised manuscript text omitted]

---

## Author Comment (AC1) · 25 May 2018

**Response to Anonymous Referee #1**

**General comments:**

This paper brings together data on glacier terminus position, speed, fjord geometry, and other metrics to examine glacier behavior across northern Greenland over 1948-2015. This is useful data to publish and results are in line with established ideas on glacier dynamics, influence of fjord geometry, and behavior of glaciers with or without floating ice tongues. Several tables' figures are particularly useful for visualizing the results (e.g, Table 2 and Figure 6) and the paper adds new information about several glaciers and is quite thorough in addressing all marine-terminate northern glaciers.

We are very grateful for this careful and constructive review of our manuscript. We also appreciate that you think the data we present is of use to the wider scientific community on outlet glacier behaviour in northern Greenland. To address the comments of both reviewers we have undergone a large re-structure of the manuscript, in the hope of significantly improving it over its previous version. We have made a real effort to simplify the structure, and focus the paper on the behaviour of glaciers in the region depending on their terminus type. Several modifications to figures have been made, while Table 2 and Figure 6 are still present in the manuscript we have added new figures to show velocity and surface elevation change through time alongside terminus change, and separate figures to display the basal topography at each glacier, more clearly than the previous centreline profile plots of bed elevation. Within the text, a large number of changes have been made when restructuring the manuscript and for that reason a tracked changes document compared to the original submission will look relatively chaotic. Several of the more specific comments outlined below may no longer be present, and several that are, lie on substantially different line numbers than in the previous version. We hope to have sufficiently signposted to all changes using the new line numbers at which these exist within a revised manuscript. Below are referee comments in black, and our responses to each comment are in blue.
* * *
Despite the strengths of the paper, there are fairly substantial areas for improvement:

**#1** In an attempt to pull climate and ocean conditions into the analysis, the authors include air temperature data from two weather stations and sea ice concentration from passive microwave (Section 2.4 and results in Section 3.5). The value of including these data seems extremely limited. On the air temperature side, only two weather stations are available, at the southern edges of the study area on the east and west coast. These data are used for a basic determination of changes in air temperature trend. For sea ice, the 25km resolution precludes analysis in narrow fjords or near the ice edge. It is well established that these data do poorly in capturing sea ice concentration at glacier termini in Greenland. Thus both the air and ocean data is severely lacking in detail compared to the other datasets the authors are working with. The authors even note themselves that they are focusing on ice dynamics and not air/ocean forcing (page 7, lines 3-5). I suggest that the authors reconsider the utility of these data and inclusion in the paper. They may instead choose to refer to data already published on Greenland air temperature and sea ice trends. The other analysis in the paper is of more interest and better quality.

On reflection, we agree with the referee's comment on the limitations of the climate data and our analysis of it in this manuscript. We also agree that it may detract from the better quality long term terminus change record and assessment of geometric controls that are instead the focus of this paper. We have taken your suggestion to remove the entire climate forcing section from both the methods and the results. Alongside this, we have shortened the first section of the discussion which referred to climate data. Instead, this section considers the timing of a switch to greater retreat with reference to previously published literature on temperature trends. We also mention some of the climate-forcing controls that have been previously considered important in northern Greenland. In line with the tidewater glacier cycle literature we have instead switched the focus of this section to climate-forcing initiating an initial terminus instability (e.g. glacier thinning), before the role of glacier geometry becomes a more important control on continued retreat and dynamic glacier behaviour.

**#2** At no point do the authors discuss some of the fundamental differences expected in glaciers with grounded termini versus floating ice tongues. I expected some acknowledgement that the former would have small, more continuous calving events and the latter would experience calving of large tabular icebergs. Since this is exactly what the authors observe, they need to provide some information and context for the behaviour. This can also include a discussion of why smaller dynamic changes might be expected for glaciers with floating ice tongues. Without some of these notes, the results and discussion feel as though they have been pulled out of context from the greater body of glaciological literature.

This is a very good point, and we realise the need to put dynamic glacier behaviour in northern Greenland, into better context with fundamental differences expected between glaciers with grounded or floating termini across the region. We have now added a paragraph to the introduction that discusses some of the cyclic behaviour expected of tidewater glaciers, and the key differences in floating and grounded terminus calving behaviour. The differences in dynamic behaviour between these two terminus types is then our justification for our new categorisation of glaciers within the region. Following some of the comments from referee 2 we simplify our previous categorisation to now just compare glaciers based on whether they terminate in a floating ice tongue or are grounded. In the discussion (Section 4.2) we reiterate that we expect the calving and dynamic behaviour of these two categories of glacier to be different, alongside explaining how our results indeed show two dominant calving patterns that are dependent on glacier terminus type.

**#3** The paper does suffer some overly complex sentences, wordy phrasing, and occasional poor organization. These items can be taken care of with mindful editing. Joshua Schimel's book Writing Science is an excellent reference for techniques and ideas.

We appreciate the advice on improving the writing of the manuscript and the book recommendation. We agree that there are sections of the manuscript where our phrasing and organisation could be improved. Throughout restructure and rewriting of several large sections of the manuscript we have paid careful attention to sentence and paragraph structure, and hope to have improved on all these factors throughout the paper. In particular these changes are within the results and discussion.
* * *
**Specific comments by page/line number:**

1/12. 'remains unknown' is an overstatement and needs changing

Changed to 'is poorly constrained'

1/23. This sentence is long and the wording at the end is overly complicated. Requires editing

1/20. We have improved the wording of the last sentence of the abstract to be shorter and more focused.

2/4. Moon et al. 2012 is a paper about ice speed and does not discuss thinning or retreat. This paper is incorrectly referenced in several places in the manuscript (e.g., also 14/32). An appropriate reference for thinning is: Csatho, B. M., A. F. Schenk, C. J. van der Veen, G. Babonis, K. Duncan, S. Rezvanbehbahani, M. R. van den Broeke, S. B. Simonsen, S. Nagarajan, and J. H. Van Angelen (2014), Laser altimetry reveals complex pattern of Greenland Ice Sheet dynamics, *Proceedings of the National Academy of Sciences*, *111*(52), 18478–18483, doi:10.1073/pnas.1411680112.

We apologise for the incorrect reference to this paper and have changed this to be Csatho et al. 2014 instead. This is now on 2/3. In the discussion (14/32) the sentence that incorrectly referenced this paper has now been removed from the manuscript as we shortened this section when condensing the climate material in the paper.

2/19-26. This paragraph would be better ordered: Sentence 2, sentence 1, sentence 3.

2/27 – 3/2 We have restructured this paragraph and to avoid overlap removed sentence 2. We now introduce the region in more detail in Section 2.1 (Study region).

4/15-19. Another section that could be simplified/shortened. For example: 'Presence of sea ice and highly fractured termini made terminus picking at Steensby, CH Ostenfeld, and NGIS glaciers more difficult (Refs). Re-digitising all 1999-2015 Landsat terminus positions yielded additional errors of ⊡13% for these glaciers.'

We are grateful for the suggestion as to how to simplify/shorten this section of the text. We have taken it as written and changed it on 5/25-28.

5/3. It's not clear what range you are referring to – include the numbers here instead of 'this'.

We have amended this to 'incrementing the number of breaks' now on 6/18

5/3-5. This is confusing and I do not clearly understand the process from this description. Please revise.

We have reworded the last couple of sentences at the end of 2.2.3 to improve the description of the process. 6/18-22.

5/11. It is better to refer to 'earlier' and 'later' instead of 'first' and 'second'.

7/6. We have amended 'first' and 'second' to 'earlier' and 'later'

5/12. Please specify what you are using to estimate average errors in velocity. This is more clear for other methods descriptions.

7/7 We have added in 'Using dataset error maps' to better describe that we calculated average errors in velocity from the included dataset error maps.

5/29-30. Why use only the difference between 1995/96 and 2015/16 velocity data to calculate change when you have so many years of data between these years. Seems that finding a trend across all years of data would provide a more accurate picture of change.

This is a reasonable point, but our terminus position changes focus on longer-term decadal changes in frontal position change. Thus, we have tried to focus on velocity changes over similarly long time-scales. It is also likely that short-term changes would be more likely to be subject to potentially stochastic variations or variations that lie within the error. Thus, we prefer to focus on longer term velocity trends and we are already conscious that the paper contains a lot of detail and is quite long.

6/9-11. The same comment as above, but for the surface elevation change. Why use just two periods when you have more data in between? As a separate note, please reconsider using 'SEC'. This is not a commonly used acronym and the more you can avoid acronyms the easier it is to read.

See response to previous point. We have also removed the acronym 'SEC' throughout the manuscript.

6/30-31. It is not clear what using 'a flow accumulation threshold of 500 to calculate stream threshold' means. Please clarify.

9/1-4 We have alleviated the confusion in explaining this method, and in doing so have removed the flow accumulation threshold of 500, as this does not affect the drainage catchment delineation.

7/30 and throughout manuscript. Remove 'clear'. This word is used widely throughout the paper and is superfluous. Recommend removing it in all cases.

9/19 Removed 'clear' on this line. We have been through and removed the word 'clear' in several places in the paper e.g. on what was 11/19 and on what was 14/5

8/8. Remove '1948-1975' from the first mention, and put these years in the second half of the sentence when you call out that the earliest epoch is 27 years long.

We have moved 1948-1975 to the second half of the sentence. Now on 10/13

9/21-31. This description is poorly organized. I want a sense of what is happening at each glacier. Separate them out and talk about each with greater specificity. Describe how advance/retreat phases were more/less consistent and then changed (or not). How has the character of terminus change varied? I understand the urge to create something of a laundry list of information, and the difficulty into crafting fairly dry information into something that is easy to follow and structured across the paragraph. It is, however, important to work towards this goal. An good example of an organized, engaging description is page 12, lines 27-32.

This section of the results has now been moved from the subheading, and most of the description moved to the floating ice tongue category. While we appreciate the comment on improving the amount of information on individual glaciers, we also feel that the paper is already very detailed in places, and are hesitant to expand on this any further. We are also aware that referee 2 has stated that there is a lot of information provided that makes it difficult to keep track of the important factors. We are obviously happy to add more detail in a further revision if needed.

10/22. 'Loss of their floating ice tongues' is incorrect for Petermann – instead just refer to 'retreat' or similar.

During the restructure this sentence has now been deleted.

10/27. Something is not 'synchronous' with events in the following decade. Reword.

This has now been changed to 'were followed by gradual glacier acceleration in the subsequent decade'. Now on 16/5-6

10/30-11/1. It's not clear if you mean changes in speed after large calving events or only after complete ice tongue removal. Please clarify.

We hope to have clarified this on 16/8-10

12/2-15. 'Dramatic' appears several times in this paragraph – it's not a particularly useful or quantitative descriptor and I recommend revising/deleting. ('Clear' also appears several times in this paragraph).

17/16-32 Removed dramatic and clear throughout this paragraph.

13/19-14/2. Another paragraph in need of reorganization.

This section of the manuscript has been re-written to improve organisation and now refers to the two terminus type categories of glacier. It also includes reference to newly included bed topography figures.

14/5 and 8. It is incorrect to refer to a single year (1995) as a change point because you are considering longer epochs. Refer to changes before/during/after those epochs rather than at specific years.

We have now changed this to 'showed a transition from slow low-magnitude advance between 1948 to 1995, to rapid high magnitude retreat between 1996 and 2015' now on 22/23

14/10. Clarify that 'These changes' is not referring to air temperatures.

We have now changed this to 'This switch to terminus retreat in the 1990s is coincident with increased air and ocean temperatures across the Greenland ice-sheet.' Now on 22/28-29

14/15. This paragraph needs an introductory sentence and work on organization and flow.

This section of the discussion has been rewritten/restructured compared to the previous version as climate-oceanic controls have become less of a focus of the paper. The second paragraph of this section now (23/8) has a better introductory sentence. It is also better organised, and summarises some of the main climate-ocean controls on outlet glacier behaviour in northern Greenland.

14/26. The second half of this sentence is irrelevant to the discussion.

This section of the discussion has been shortened, and describes climate-ocean forcing factors in far less detail with no real evidence in the results presented in this paper. This sentence has been entirely removed.

15/2-3 (and following paragraph). Acknowledge the role of other ocean processes, like ice front melt, in this sentence/section, followed by the more thorough discussion in the next paragraph. These references (or information within them) may be useful:

Wilson, N. J., and F. Straneo (2015), Water exchange between the continental shelf and the cavity beneath Nioghalvfjerdsbræ (79 North Glacier), *Geophys Res Lett*, *42*(18), 7648–7654, doi:10.1002/2015gl064944.

Choi, Y., M. Morlighem, E. Rignot, J. Mouginot, and M. Wood (2017), Modeling the Response of Nioghalvfjerdsfjorden and Zachariae Isstrøm Glaciers, Greenland, to Ocean Forcing Over the Next Century, *Geophys Res Lett*, *44*(21), 11,071–11,079, doi:10.1002/2017GL075174.

As we have now removed the climate data presented in this paper, we have largely removed the discussion on climate-ocean forcing. This is partly following the comments from referee 2 which suggested that by discussing so many different controls (climate, topography, terminus type), it was difficult to determine the main factors/focus of the paper. One of their comments was also that some of these climate-ocean processes discussed in the previous version of the manuscript were being 'invoked…with little evidence'. In an attempt to refocus the paper, we focus on terminus type, and glacier geometric controls. We include some comments on climate-ocean forcing (Section 4.1) but this is mainly with the direction that climate-ocean forcing may have changed the initial conditions at the terminus, but after that, terminus type and geometry are the main controls on the different behaviour of outlet glaciers in northern Greenland.

16/2. Write these in an order than makes more sense for the actual process, either thinning- retreat-speedup or retreat-speedup-thinning (use this latter one if you want the focus on dynamic thinning due to speedup).

We have changed this section slightly to make more reference to the calving styles and discuss the effect on glacier force balance. However we have re-ordered these processes on 23/33 to 'thinning is thought to have initiated enhanced retreat and accelerated terminus velocities'.

16/15-19. It would be useful for the authors to comment on why they think these differences occur among the glaciers they mention. For example, how does scale of event and force balance based on glacier characteristics enter into the discussion. Also, it's not entirely clear whether the authors are consistently referring only to velocity changes on the grounded ice portion of these glaciers.

We are grateful for the suggestion to add in some discussion of how the scale of calving events and differences in the setting of each glacier affect the force balance and thus differences in glacier dynamic behaviour. In this section of the discussion and the following (Sections 4.2 and 4.3) we have made an effort to address this point and make more reference to the calving style of these two terminus types of glacier, and how the differences could impact on the force balance. An example of this is on 24/17-19 where for floating ice tongue glaciers we say: 'However, in most cases large calving events, appeared not perturb the force balance by neither increasing longitudinal stretching, nor driving stresses on inland grounded ice'. We go into the impact of such calving events and the fjord setting of each glacier in Section 4.3. We also discuss there the different forces (basal vs lateral drag) acting on grounding or floating termini. For example on line 25/28-29 in reference to floating ice tongues we say 'lateral resistive stresses are the main control on the glacier force balance and driving stresses'. We hope this has now made it clearer throughout the impacts of the scale of calving events and alterations to the force balance at these two types of glacier. In response to the second point, our newly created figures for terminus, velocity and elevation change, we include velocities averaged at the grounded line region of each glacier.

17/4. Another paper just out on this topic: Millan, R., E. Rignot, J. Mouginot, M. Wood, A. A. Bjork, and M. Morlighem (2018), Vulnerability of Southeast Greenland glaciers to warm Atlantic Water from Operation IceBridge and Ocean Melting Greenland data, *Geophys Res Lett*, 1–23, doi:10.1002/2017GL076561.

Included this reference, now on 25/11.

18/4 and 11. What do the authors mean by 'strongly attached to'? How has that been quantified, in this study or others?

26/3-5. We have changed this sentence to now read 'In particular, C. H. Ostenfeld and Hagen Bræ, have heavy rifting along their shear margins, appear relatively un-confined by their fjord walls, and weakly attached to the grounded terminus (Figure 11b,c)'. This now makes reference to a new figure we include that shows satellite imagery of the ice tongues of these glaciers before they collapsed.

19/18-19. A few more words are needed on this, and whether or not it is likely these are surge glaciers. Did you look at different data than these other studies? Can you definitely confirm there was no surge in periods where it was previously detected because you have better data or similar?

27/31. We have added in an extra couple of sentences to explain the observations made by the studies on these glaciers to suggest surge behaviour. We also add how our long-term record, where we consider terminus changes alongside elevation and velocity changes, provides no substantial evidence for surge-activity at these glaciers. We are wary of adding too much more detail here, as these glaciers are not the main focus of this section, and we instead want the majority of the discussion to be about those which have substantial evidence.

19/20. 'controlled by external forcing' is too vague. Say specifically what mechanisms might be at play and whether there is evidence for it, or what data would be needed.

This sentence has been removed, and we instead just state that while there has been a large advance (similar to some previous observations) we lack detailed data to be able to provide more substantial evidence of it being surge-type.

19/30. Another incorrect reference to Moon et al. 2012. This would be a good place to reference Howat and Eddy 2010 (already listed in the references).

Removed Moon et al. 2012 reference and replaced it with Howat and Eddy 2011.

20/2. A variety of ocean data is available for northern Greenland. It is not, however, being used or analysed in this paper (which is just fine). But please remove this incorrect statement.

This sentence has been removed.

20/24. I understand the urge to end on 'could soon contribute an important component to sea level rise', but this is a vague statement and is not well connected to the paper analysis (which does not discuss sea level). Suggest rewording with a stronger concluding statement that is more specific and tied to the main idea of the paper.

We have removed this sentence and now conclude the paper with a stronger concluding statement that focuses on the main findings of the paper, e.g. region wide increase in retreat rates, differences between terminus types, and the important role of glacier geometry. We also highlight that while ice tongue retreat doesn't appear to matter, once these glaciers become grounded they may discharge greater volumes of grounded ice to the ocean.

27/3. This caption would benefit from more precise language throughout. The use of 'calculated by subtracting 1948 and 2015 positions' is one example.

We have improved and shorted the caption for this table.

Table 1. Consider the various order in which glaciers in each category could be listed and choose the one that makes the most sense for the reader or message.

We have now changed the order within each terminus type category to be based on frontal position change rate from highest overall (1948 to 2015) retreat rate through to the highest advance rate. We have used this same ordering for all figures that follow in the manuscript.

Figure 1. The caption includes a lot of information on methods, which seems misplaced.

We are not sure about this comment, there does not appear to be much detail on the methods in this caption. However, in the caption of Figure 2, we do refer to the methods too much, and we have reduced this to just describe the figure.

Figure 2. The legend should have lines rather than boxes.

We have updated the legend to now show lines for terminus positions rather than boxes.

Figure 4. Please reword for improved clarity and brevity.

We have improved the wording of the figure caption for Figure 4.

Figures 7-9. It is very difficult to see the lines/colors in the legend and in the plots. Distinguishing among the surface elevation change lines to understand their progress is only possible in a broad green or blue sense. Understanding the detailed progress is impossible with the current color map.

To improve the presentation of surface elevation and velocity change over time we have replaced figures 7-9 with two figures that are categorised based on terminus type, and show individual glacier frontal position changes and average elevation change along the centreline profile (due to poor resolution at the terminus) and average velocity at the grounding line. We hope this has now significantly improved the ability for the reader to understand the detailed progress of elevation and velocity change alongside terminus changes over time.

Figure 8. Remove the odd floating ice in 8h, which does not appear to be connected to the glacier.

This figure has now been removed and new figures showing bed topography profiles have been created (Figs. 9 and 10). This section of floating ice does not appear on any figure in the manuscript anymore.

Figure 9. Is there no data for showing terminus position in 9c?

This figure has now been removed from the manuscript. On the newly created bed topography Figures (9 and 10) terminus positions are shown for all glaciers.

Figure 10. Instead of 'inland' and 'terminus' give a number for actual location/distance.

We have also replaced this figure to include one that shows the original profile surface elevation and velocity data (Figure 8). We feel that the colour scale here can provide a clear representation of the overall trend of increased elevation inland, alongside reduced velocity, compared to velocity increase and thinning at the terminus.

Figure 11. It's quite odd to stack the warmer temperatures below the colder temperatures in these plots. You also mention 'ocean' in the caption data, which is not included in the plots.

As we have taken the advice to remove the climate-ocean forcing section of this paper due to poor data quality, this figure is no longer included in the paper.

**Technical corrections by line number:**

2/5. Delete 'across the ice sheet' – unnecessary.

Deleted.

2/30. Delete 'objectively' – unnecessary.

Deleted.

7/30. Delete 'eventual'

Deleted.

10/8. Delete 'It was also clear that'. I'm not going to note anymore of the instances of 'clear', but just repeat that they should all be removed.

Deleted. We have also been through the entire manuscript and deleted all instances of 'clear'.

11/19. Thickening or thinning?

This has now been changed to 'Small increased thinning or reduced thickening rates at Academy and Marie-Sophie Glaciers (1999 to 2000: Figure 6f,h)'.

11/21. Delete 'then'

Deleted.

15/7. 'concentrations' instead of 'conditions'

When removing the climate section of the paper, this sentence has also been removed.

15/9. Remove quotes around calving season.

When removing the climate section of the paper, this sentence has also been removed.

15/11. Remove ','

When removing the climate section of the paper, this sentence has also been removed.

17/19. 'importantly influence' is very awkward – reword

When restructuring this section of the discussion this sentence has been removed.

18/24. Should be 'accompanied by acceleration'

The discussion of Ryder Glacier has been changed (following the advice of referee 2) and so this sentence does not longer exist in the manuscript.

19/17. 'overriding' is poor word choice – please change

Changed to altering

35/5. Replace 'Current' with '2016'

This has been changed in the captions of newly included bed topography figures to 'the most recent recorded terminus position (2015) from this study'.

Figure 12. Delete 'except for the first…position changes

This figure has been deleted.

---

## Author Comment (AC2) · 25 May 2018

**Response to Anonymous Referee #2**

**General comments:**

This manuscript presents a large volume of data for several glaciers in N. Greenland to draw conclusions about their dynamic behaviour over a significantly long time period. The data presented are of some value, but I'm afraid that this manuscript suffers a bit from explaining everything without really explaining anything. What I mean is that there is a dizzying array of information about climate, topography, glacier behavior to keep track of but not one factor comes across as being important to explaining the behavior of all glaciers. It's a challenge for the reader to keep track of all of the information and to make sense of what facts are important throughout the text.

We are very grateful for your detailed and constructive comments on our manuscript. We have undergone a major restructure of the manuscript in the hope of improving it substantially from its previous version. In doing so, we hope to have refocused the paper, removed some of the excess information, and improved the readability. While there is a lot of information put forward, to some extent this region is a complex one, and cannot be explained by a single factor alone. That said, throughout we have tried to focus more on the role of terminus type in controlling the differences in behaviour in northern Greenland, and the influence of topography. Following the advice of referee 1, we have also entirely removed the climate forcing section, to focus on terminus type and topography. We also feel that by improving the categorisation of glaciers in the region (see next response), we have removed a large amount of the confusion, and highlighted the important factors that we focus on in this paper more clearly. Despite the volume of data that we have presented, we do feel that it is of value to the scientific community, as it provides a new record of outlet glaciers change in northern Greenland. As well as this, referee 1, did point out that in some cases there needed to be more detail on specific glaciers, and so in several places we were cautious about removing any of the detail on individual glaciers.

The other issue I have is with the categorization of glaciers. There are categories 1) grounded terminus; 2) floating ice tongue and 3) potentially surge-type. Two of these reflect the state of the terminus while the last one reflects the inherent dynamics inferred from terminus behavior. Further, several of Category 3 have (or had) floating tongues, making it a challenge to keep up with the author's thoughts at times. Then, there is a second categorization – based on retreat style: 1) steady retreat; 2) rapid retreat; 3) advance. It's just too much to keep track of. In the conclusions, the authors say that "a key conclusion is that the dynamic response of outlet glaciers to perturbations depends on their terminus type". However, with such poorly organized material and categorization it's unclear how this conclusion is supported.

We appreciate your feedback on the categorisation of glaciers within the manuscript, and this is one of the major aspects we have tried to address in the revision. We agree that there was confusion before, and to alleviate this, we have decided to categorise solely on the terminus type of the glacier in northern Greenland. We still include the results of the changepoint analysis to provide evidence for the differences in the duration/magnitude of terminus changes for these two categories, but do not use it to objectively categorise the glaciers initially. The changes we have made to restructure based on these two categories of glacier based on terminus type are summarised in the following sentences. Firstly, in the introduction we give context to the expected differences in glacier behaviour between glaciers with grounded or floating termini. Then, we have added a new study region section at the beginning of the methods, where, as suggested, we provide an overview of northern Greenland, and identify which glaciers are grounded at their terminus, which had ice tongues over the last two decades (1995 to 2015), and which glaciers in the region still terminate in a floating ice tongue. Then at the beginning of the results Section 3.2 we reiterate these changes, and state that we consider these two categories of terminus type throughout the manuscript. Each section of the results has been restructured and largely re-written accordingly, to be categorised by these two terminus types, and include glaciers with floating ice tongues that were previously in the 'sustained advance/surge-type' category. In the discussion we leave the structure as it was, including the 'Glacier Surging' section, but instead suggest surging at those with the most substantial evidence, and provide an alternative explanation for the behaviour of Ryder Glacier. We hope this now provides a much simpler structure to the results and discussion by comparing these two categories based on terminus type. We think this the manuscript is a lot clearer, and hope this will be the case for the reader too.

In addition, the category of "sustained advance" is completely untrue. These glaciers (shown in 6c) undergo periods of advance AND retreat, in some cases, very rapid retreat, which is a far cry from sustained advance. This type of behavior

is typical for glaciers with floating tongues – see MacGregor et al., JGlac 2012 58(209) and other tidewater glaciers – see McNabb et al., JGR-ES 2013 for additional examples of this. I feel as though categorizing these glaciers as "surge type" is a bit "getting off too lightly"– there is likely more to explain here. The authors would benefit from a close read of Steiger et al. (Cryosphere 2017) that suggests pinning points having an impact on glacier terminus positions. Also, the authors should examine the tidewater glacier cycle literature which discusses quite broadly the idea of cyclic glacier changes.

We have largely addressed this point in the previous comment. We also agree that sustained advance is not true at many of these glaciers, and understand the importance of considering these within the tidewater glacier cycle literature. We also agree that there is more to explain here, without jumping to the conclusion of glacier surging. Through our re-categorisation based on terminus type (grounded or floating) alone, we have removed the final category which covered glaciers which had undergone periods of 'sustained advance'. Three of these glaciers have floating ice tongues, and so we discuss these alongside other floating ice tongue glaciers within each section of the results. We have left the discussion section that discusses surging at some of these glaciers, and in the case of Ryder Glacier, we have provided a clearer explanation for its cyclic behaviour based on the evidence provided in the paper.

The authors use BedMachine v2, when BedMachine v3 has been released now for a year. v3 represents significant improvements, particularly in the terminus regions because of the addition of bathymetry data from the OMG project. It would be useful to know how the authors determined if the bed data were good or not. Some glaciers were not sufficiently sampled with radar data for the mass-conserving solution and thus, are not well-constrained in BedMachine. Finally, very little information is provided about how the authors calculated bed slopes at the glacier termini and how bed topography is used in general. The mention of pinning points and the comparison between slopes of the beds of glaciers is described with no data presented.

To our knowledge the latest version of BedMachine v3 was only realised online last September, by which time we had completed the majority of our analyses. We do agree that there are significant improvements in this dataset and so we have spent time going back and incorporating v3. This includes a much more detailed section in the methods, where we consider the errors in the dataset at each glacier in northern Greenland, and describe the method by which we calculated glacier bed slope direction. We have also provided more detail in the supplementary information on the errors and bed slope direction. Additionally, we have created new figures to replace Figures 7-9 in the previous version of the manuscript. This includes figures for each terminus type that show terminus change and elevation/velocity changes averaged along the profile/at the grounding line (see specific comment below), and new figures to show the basal topography of each glacier more clearly. We use individual plots to show the spatial bed topography of each glacier, and below the bed elevation along each glacier centreline. This also includes the location of the current terminus as included on the previous versions of bed topography figures. We hope to have now made a much clearer assessment of the bed topography beneath these glaciers, and a clearer explanation for our method of bed slope calculation.

In the discussion, the authors invoke processes such as increased ablation rates, water drainage to the ice bed, and the removal of sea ice to explain the timing of glacier retreat. However, these correlations are presented as anecdotes, with very little in the way of evidence suggesting cause/effect. They discuss topography as well stating that bed topography is a "key control on the behaviour of glaciers in northern Greenland" but provide very little in the way of evidence for the reader to understand how this conclusion came to be. Bed topography is inherently three-dimensional and so presenting the topography in the small-scale images in the figures is not sufficient evidence for the reader.

In response to the first sentence, we have followed the suggestion of referee 1 and removed the climate data from the manuscript. This includes removing the section of the discussion that covered such processes of ablation, and water drainage. The main focus of this paper is not to explain these processes in any detail, and we agree that we do not have the evidence in the data we present to support these suggestions. Instead we have shortened the first section of the discussion (4.1) which covered the timing of glacier retreat, to now refer to previously published literature, which has highlighted several processes that may be important in forcing glacier retreat in northern Greenland. We adjust the focus from suggesting which processes may have controlled the timing of retreat, to the idea that climate-ocean forcing

may have been the initial driver of rapid retreat, after which glacier geometry becomes a more important control on glacier retreat.

In response to the comment on bed topography, we have restructured Section 4.3 of the discussion which covered topographical controls on glacier behaviour. We hope that by including more detailed figures of the bed topography at each study glacier, we have made the evidence clearer for how the glacier geometry can be a control on glacier behaviour. As well as this, we have re-written this section (4.3), to focus more heavily on how the force balance may have been altered by the glacier geometry, and also included a figure (11), which gives examples of the confinement of three ice tongues in satellite imagery, to support our argument on the differences between lateral resistive stresses at these ice tongues.
* * *
Some additional edits are made in-line with the text in the attached pdf, but towards the middle I stopped correcting small things.

It would be nice to have a paragraph here (at the beginning of the methods) describing the region and which of the glaciers have ice shelves etc.

As mentioned above, we appreciate the advice to add a paragraph that introduces the region. We have done this, by giving an overview to the region we define as northern Greenland, and included a description of which glaciers have floating ice tongues. We have also included symbols on Figure 1, to show which glaciers currently have floating ice tongues, which have recently lost them (1995 to 2015), those which have some historical evidence in the literature for floating ice tongues, and those which are grounded at their terminus throughout the study period.

3/9 Am I correct then in assuming that just one image per year was used in analysis?

Yes and we have updated this in the manuscript.

6/21 this reads as if the authors performed this work, but I am assuming that they merely used the existing bed map derived by Morlighem. Also, there were significant updates in BedMachinev3 that should be incorporated.

We have updated this to read less like we did the work ourselves. As stated in more detail above, we have now incorporated the newest version (v3) of the BedMachine dataset

6/28 refer in here to Table 2, where these data are presented

We have added in reference to Table 3 (formerly table 2 as we have added an additional table showing mean decadal terminus changes for each terminus type).

7/4 RACMO and MAR data provide basic climate data for all of Greenland. Instead, I would just eliminate this sentence.

This entire climate section has been removed, and along with it, this sentence.

9/22 except for Ryder, which you show advancing as early as 1950.

We have removed this sentence to avoid confusion.

10/2 which ones are floating?

We have removed this subheading entirely during the restructure of the categories, so the subheadings are now split by grounded-terminus outlet glaciers and floating-terminus outlet glaciers

13/2 This has resulted in a noted increase in glacier runoff published by Brice Noel in 2016 (I think).

This climate section of the results has now been removed entirely.

17/11 I doubt that retreat rates would be related to the slope of the over-deepened bed, but instead to the balance of forces at the glacier terminus.

When restructuring this section of the discussion we have removed this sentence, and focus more generally on the impact of deep basal troughs on the stability of the terminus.

**Figure 6:** I'm confused by this categorization into a), b), c) because Fig. 9 shows that several of the "potential surge-type glaciers" also have floating ice shelves

We have changed the categorisation throughout the paper, and therefore have adjusted this figure to just show periods of minimal and rapid terminus change. The glaciers have then been ordered based on terminus type, either floating or grounded terminus, and by overall frontal position change rates (Table 1), within each category.

**Figure 7,8,9:** it would make more sense (to me) to see the terminus-averaged velocity plotted with time to compare to the time-series of the terminus position.

We agree and have made substantial changes to these figures to show this data. Figures 7-9 have been replaced by two figures (Figure 6 and 7) which show terminus changes with time alongside grounding line averaged velocities, and elevation changes averaged along the entire glacier profile (due to poorer resolution). We believe this new figures provide better representation of the results where we discuss changes in velocity and surface elevation change through time. These new figures improve the visualisation over the previous figures which were difficult to see specific changes in elevation/velocity through time.

**Figure 12:** Since you refer to NEGIS in the manuscript when referencing this figure, it might be good to add glacier names to it.

As we have removed the entire climate section from the manuscript this figure has now been removed.

---

## Referee Report (RR1)

This study combines glacier terminus position, velocity, surface elevation and bed topography datasets to investigate glacier dynamics across northern Greenland between 1948-2015. The paper nicely presents both long-term trends and regional variability based on terminus types (floating vs grounded) and local fjord geometry. The paper would benefit from a greater and more concise focus on these points. Ultimately, despite some useful insights, there remains several areas for significant improvement. I've also included specific page/line number comments below.

**1 Only including velocity changes between 1995/96 and 2015/16 might alias important velocity changes on shorter timescales that could be linked to discrete terminus perturbation events. As such, the link between terminus position and dynamics might not be fully appreciated. Perhaps finding trends across all years would provide a more complete context and links to the terminus position changepoint analysis? The same could be said for surface elevation changes; why not look at shorter-term trends?**

**2 The introduction of terminus types – grounded or floating – is a great distinction and worthy of investigation. However, it should be made explicit up front and not part way through the results. Furthermore, I find it hard to follow the results section for frontal position change. What is the main point you want to make? It seems redundant to go through so many different periods and classifications of change; net from 1948-2015; decadal; changepoint time periods; based on terminus type. I would change to 1) briefly note trends and variability over the entire study period, 2) introduce terminus types (grounded vs floating), and 3) differences in frontal positions between terminus types at decadal (i.e. fig. 4) and/or changepoint time periods (i.e. fig. 5).**

**3 The discussion introduces several triggers for enhanced terminus retreat, including "initial thinning at the glacier terminus." While possible, I do not think that these suggestions are well supported within the data analysis and results. The authors note in the methods section that thinning rates were averaged over the entire glacier centerline, so do we have the spatial resolution to test this hypothesis? Does retreat lag thinning in the time series? Is thinning dynamic or SMB driven? Furthermore, if large thinning rates cannot be explained solely by SMB, wouldn't terminus retreat be required to produce the observed thinning rates? The authors present a multitude of descriptive data in the results section, however, I feel there are gaps in logic within the discussion in attempting to explain the observed trends.**

**4 Throughout the manuscript the authors invoke climate forcing as a possible trigger of terminus retreat and dynamic glacier adjustments, however, the authors do not include time series of climate and ocean conditions. I certainly appreciate that climate forcing is not the main focus of the study, but perhaps it is worth including some available data in the supplementary information for readers and reviewers to look at. If not, the authors should consider more careful and direct references to pertinent published datasets and studies**

**5 Where is the calculation of the force balance, longitudinal stretching and driving stress that is referenced in the discussion? Please make it explicit if we are supposed to deduce these from the velocity time series alone.**

**6 Is the discussion of surge-type glaciers relevant to the main conclusions of the paper? It seems to confound the main points: behavior of grounded vs. floating termini and importance of bed topography controls.**

**7 Perhaps most important - the manuscript writing should be more clear and concise. The main point within individual sentences or paragraphs is often convoluted and, as a result, the content suffers significantly. I've tried to offer some specific improvements in my line edits, but was unable to address everything. Ultimately, these problems can be addressed with careful and collaborative editing by all authors.**

**Specific comments by page/line number and figure number:**

1/16. No need for parenthesis, just "was"

1/19. "adjustment" not "re-adjustment"

1/21. Delete comma before suggests

1/29. Should be Carr, 2017a

2/5. Delete "surface"

2/10. This paragraph is longwinded – considering stripping down to the main points, i.e., terminus retreat can initiate dynamic adjustments independent of climate and modulated by local outlet geometry and associated resistive stresses. The last two sentences seem most important.

2/12. I suggest using "slow", "long", or "gradual", but best not to use two adjectives.

2/17-19. Is this sentence necessary? If so, perhaps it should have a reference.

2/27. Delete "Most"

2/30. Create a new sentence…"Dynamic changes at Jakobshavn are linked to the gradual collapse of its floating ice tongue."

2/31. Is there anything specific that can be added here to demonstrate the importance of northern Greenland ice dynamics to sea level rise? Important to let the reader know the region is important to study for reasons other than it's underrepresented in previous investigations.

2/33. Delete "far"

3/1. "Consequently, few long-term records of frontal positions exist in the region. As a result, their potential impact on inland ice flow remains unclear."

3/5. The sentences in this paragraph seem redundant. I would suggest combining sentences 2-5 into something like, "We couple a multi-decadal annual terminus position record between 1948 and 2015 with recently published surface elevation and ice velocity datasets. We use these datasets to evaluate dynamic responses (i.e. acceleration and thinning) to frontal position change and examine disparities in the context of glaciers with floating or grounded termini."

3/10. Would recommend changing slightly to, "Finally, we assess local topographic setting (ie fjord width and depth) as a control on glacier behavior."

3/16. Is this true? There are other, albeit smaller floating tongues elsewhere, such as Rink Isbrae and Helheim?

3/13-20. This seems like introduction or nonessential methods material. What is the point of this paragraph? Seems like most important information is the characterization of floating vs grounded termini... then quickly note that there are large and changing tongue systems.

5/2. Are there any gap years?

6/7. To what end? Do you use changepoint analysis between glaciers, over a single record, etc.? What is the point? This paragraph needs a topic sentence that makes this clear up front for the reader to understand the value in this approach.

6/10. This sentence is redundant and could be more concise. Just cite Bunce and Carr in the first sentence after clarifying.

6/18. Within what range?

6/17-22. This explanation is confusing to the reader. What is the reason for a threshold penalty? What is a threshold penalty? If this method is following Carr 2017, then simply reference their method, give a brief overview with an emphasis on portraying what the main point is and why it's valuable. The main point seems to be articulated in the last sentence of the paragraph, perhaps this could be a topic sentence?

8/9. It is unclear why Euclidean distance is necessary – to draw centerlines? What if fjord walls are not parallel?

8/11. I would think averaging elevation change over the entire centerline (to the ice divide as the manuscript suggests) would significantly skew your results. Would it also be better to mask elevation changes seaward of the grounding line on floating tongues?

9/4. This sentence is unclear – you're calculating catchment areas from the flow field right? Could you instead reorient the sentence as, "We calculated each drainage area using catchments constrained by gradients in the DEM"...?

9/7. Perhaps "Net retreat"?

9/7. Do these statements pertain to frontal positions (ice tongue fronts and grounded termini), or just grounding lines? Please clarify.

9/14. Is "mean rate of terminus change" more accurate?

9/17. Could you be more direct and just say, "Long-term retreat rates varied across northern Greenland?"

12/2-6. The distinction of terminus type needs to be made earlier to give the reader context to interpret records of terminus front change.

12/12. Already stated previously. Need better topic sentence; why do grounded termini matter? State main result up front and then support with observations.

12/23. Already stated previously. Need better topic sentence; why do floating termini matter? State main result up front and then support with observations.

17/16. Higher with respect to what? Need to clarify.

18/1, Perhaps change to, "different pattern of elevation change compared to the rest of the region: Storstrommen and L. Bistrup Brae."

19/1. Perhaps it is best to also explicitly separate this section into grounded vs. floating termini?

19/4. Please clarify what is meant by "split"

19/2 and 22/1. What are the main points of these paragraphs? Please upgrade the topic sentences to better reflect the main point – inland sloping beds are correlated with higher retreat rates at glaciers with grounded termini, but fjord width is not a main determinant. What is a corresponding main point you are trying to get across in describing bed data at glaciers with floating tongues?

22/30. Delete acceleration and retreat in southeast Greenland, or support with references and include other regions where this occurred (e.g. SW Greenland, Howat et al., etc.).

22/31. Please provide more complete and recent references (e.g. Felikson et al., 2017). Also please more explicitly link these changes to climate factors, if that is in fact what you mean to

do.

23/2. Do you show this relationship in the text? If so cite a figure. Also please reword the sentence. "Indeed, surface thinning preceded rapid terminus retreat at many northern Greenland glaciers (Fig. x)." I would be wary of adding sentences referencing old studies in other parts of Greenland and focus more on the region of interest. What is the main point you are trying to make? Is it that climate may be the initial trigger for terminus change? Then make it explicit and cut the fat.

23/2. Confusing sentence, please reword. Not initial condition, do you mean forcing?

23/9. Delete "the"

23/8. I'm confused, is it climate and ocean forcing at the terminus; or thinning from negative mass balance that causes retreat? I realize they are related (i.e. climate driven), but I think you can be more direct.

23/12. This seems like material that could be combined with the previous paragraph. In fact, I would consider making this section a single paragraph that is more direct and punchier. Lots of material is repeated and the topic lacks focus.

23/15. Can also add: Catania G. A., Stearns, L. A., Sutherland, D. A., Fried, M. J., Bartholomaus, T. C., Morlighem, M., et al. (2018). Geometric controls on tidewater glacier retreat in central western Greenland. Journal of Geophysical Research: Earth Surface, 123, 1–14, https://doi.org/10.1029/2017JF004499

23/25. What are the significant differences?

23/26. The last sentence of this paragraph is essentially the same as the first. Consider revising.

23/31. What causes the initial near-terminus thinning? Is this supported in the results section? Does retreat lag thinning? If so, reference the appropriate figure. Otherwise, I feel this is unfounded.

23/33. Ice does not flow inland; perhaps you mean the dynamic response propagates inland, or up-glacier?

23/33. Change to, "As such, we suggest thinning initiated enhanced retreat…"

24/19. Awkward sentence, consider rewording.

24/19-20. Where do you show calculation of the force balance, longitudinal stretching and driving stress? Also, you may not have the temporal resolution in the velocity dataset to resolve short-lived dynamic adjustments from individual calving events.

24/27. Delete the comma in this sentence.

28/10. It doesn't really seem that any glaciers are responding linearly to climate forcing. It looks like Ryder has a very nice, episodic advance/retreat cycle.

28/10. It is hard to compare these records to climate forcing without also seeing time series of climate forcing (i.e. air or ocean temperatures).

29/9. But didn't the results section showed fjord width had little control over retreat rates?

Figure 6 and 7. It's hard to read and interpret yellow colored velocity data. I suggest changing to blue or another easily accessible color.

Figure 9 and 10. The bathymetry colorbars are not needed for each panel. Consider including one at the top or bottom of the figure.

Figure 11. Please annotate the ice flow direction and make explicit the location of ice tongues and lateral rifting. It's hard to make out these features without descriptions in the figure.

---

## Author Response (AR2)

**Dear Editor,**

We are grateful for the additional round of comments on our manuscript from both yourself and an anonymous referee. These comments have helped to improve several sections of the manuscript. In particular, we feel that the focus of the manuscript is far clearer. Below we respond to the major comments by the referee, in each case including and noting the editor's guidance on each matter, as well as the specific line edits suggested by the referee. We also include a revised manuscript with changes tracked. Once again we thank the editor and the additional third referee for their constructive comments on the manuscript.

Thank you for considering this manuscript for publication.

Emily Hill

Response to major comments made by anonymous referee 3 (in each case including the moderated revisions by the Editor)

**Referee** #1 Only including velocity changes between 1995/96 and 2015/16 might alias important velocity changes on shorter timescales that could be linked to discrete terminus perturbation events. As such, the link between terminus position and dynamics might not be fully appreciated. Perhaps finding trends across all years would provide a more complete context and links to the terminus position changepoint analysis? The same could be said for surface elevation changes; why not look at shorter-term trends?

**Editor** #1 This point on considering more detailed temporal trends/resolution of velocities and elevation changes has already been risen in the first round of reviews and explained but the authors why they did not change this. I agree with the re-reviewer that for the explanation of some of the dynamic changes the detailed timing of acceleration or thinning is crucial (see major comment #3) but I understand that calculating continuous rates is with the given dataset very challenging. I suggest at least in the discussion to include a more detailed/temporally resolved consideration of temporal variations (even though in the presented spatial variations of rates (in figures and tables) this is not explicitly done).

Our initial reason for including long-term trends in velocity and surface elevation change (Table 1) was to enable comparison to our longer-term decadal changes in frontal position, and provide a broad overview of any major changes. We agree that by not providing a continuous record there may be some oversimplifications in our interpretation of short-term timescales. However, the focus of the manuscript is on the long-term dynamic changes at outlet glaciers in northern Greenland, rather than a short-term analysis of individual calving events and velocity changes. Indeed, this may even require seasonal resolution for velocity and elevation change datasets, which are not available. We agree that short-term velocity/elevation changes are likely to be relevant and we present data on shorter time-scales (Figures 6 and 7) and briefly discuss these in the revised manuscript, as the Editor suggests. However, we refrain from a lengthier detailed analysis simply because our main focus in this paper is on longer-term trends from a broad sample of glaciers. Undoubtedly, future work could look at this on some of these selected glaciers, but it would be difficult for us to do this across all glaciers in a clear and consistent manner and without adding substantially to the length of an already long manuscript.

**Referee** #2 The introduction of terminus types – grounded or floating – is a great distinction and worthy of investigation. However, it should be made explicit up front and not part way through the results. Furthermore, I find it hard to follow the results section for frontal position change. What is the main point you want to make? It seems redundant to go through so many different periods and classifications of change; net from 1948-2015; decadal; changepoint time periods; based on terminus type. I would change to 1) briefly note trends and variability over the entire study period, 2) introduce terminus types (grounded vs floating), and 3) differences in frontal positions between terminus types at decadal (i.e. fig. 4) and/or changepoint time periods (i.e. fig. 5).

**Editor** #2 I would also welcome a clearer introduction of 'terminus types' in the introduction which would allow a more focussed and clearer presentation in the results.

A discussion of the terminus types already exists in the introduction (lines 15 to 21) and we also introduce which glaciers have floating ice tongues or are grounded in Section 2.1 (Study region). However, we have followed the advice of the referee comment and restructured the first section of the results. We now have a brief overview at the start of Section 3.1 of region wide long-term changes. Then we have brought the introduction of terminus types (from Section 3.2) before the decadal changes. We agree that this gives better context for interpreting the decadal changes that are presented in Figure 4 and Table 2. We have left the decadal discussion in this section, rather than splitting by terminus type, but we have added additional focus based on terminus type in this paragraph. The section is concluded with a summary sentence that was previously at the end of Section 3.2, and states that 'based on terminus type, we treat these as separate categories for the remainder of the results, during which we discuss short-term trends'. Finally, we have removed Section 3.2 as it was surplus and was largely brought into this section, and instead go straight into individual sections on grounded and floating termini. We have changed the numbering of the subheadings accordingly.

**Referee** #3 The discussion introduces several triggers for enhanced terminus retreat, including "initial thinning at the glacier terminus." While possible, I do not think that these suggestions are well supported within the data analysis and results. The authors note in the methods section that thinning rates were averaged over the entire glacier centreline, so do we have the spatial resolution to test this hypothesis? Does retreat lag thinning in the time series? Is thinning dynamic or SMB driven? Furthermore, if large thinning rates cannot be explained solely by SMB, wouldn't terminus retreat be required to produce the observed thinning rates? The authors present a multitude of descriptive data in the results section, however, I feel there are gaps in logic within the discussion in attempting to explain the observed trends.

**Editor** #3 improve and clarify the argumentation and line of thought for the discussion of the causes of enhanced dynamic changes (retreat). Relating to point #1 the more 'continuous' time series of thinning and velocities could be better used in discussion.

We appreciate the concerns about the current dataset and absence of climate forcing for inferring the specific triggers for enhanced terminus retreat. We also appreciate that by averaging over the centreline that we cannot comment on spatial elevation changes and instead can only comment on the general trend of annual changes in thinning and acceleration shown in Figures 6 and 7. We try to make sure that the focus of these sections of the discussion is instead on the dynamic changes (acceleration and thinning) during periods of rapid retreat. For this reason we have removed any reference to 'initial thinning at the termini' from the discussion. We also appreciate the concerns that we do not have the spatial resolution of elevation change data to determine if it is dynamic of SMB driven. We have responded to specific line comments that have helped address this comment (e.g. comments to lines 23/2 and 23/31) as well as some confusion about the extent of the glacier centreline (in the methods). Following the Editor's suggestions on

Section 4.1 we have improved the argumentation for thinning rates being the initial forcing for dynamic retreat, by referring to previous studies that have invoked dynamic thinning at glaciers in northern Greenland (Khan 2014, Pritchard 2009) and have made better references to the figures that show the more continuous time series of thinning and retreat (Figures 6 and 7). In addition, by following specific comments by the referee (23/31), we have revised Section 4.2 paragraph 2 in which we also previously invoked the cause of terminus change. Instead we refer back to Section 4.1, briefly mention that dynamic thinning may have initially caused retreat, but add a caveat to our coarse resolution dataset. We then clearly focus the rest of the section on changes in velocity and thinning after the onset of rapid retreat (no matter what the cause).

**Referee** #4 Throughout the manuscript the authors invoke climate forcing as a possible trigger of terminus retreat and dynamic glacier adjustments, however, the authors do not include time series of climate and ocean conditions. I certainly appreciate that climate forcing is not the main focus of the study, but perhaps it is worth including some available data in the supplementary information for readers and reviewers to look at. If not, the authors should consider more careful and direct references to pertinent published datasets and studies

**Editor** #4 this is a tricky point, as the authors removed the extensive presentation and discussion related to climatic and oceanic forcing in order to address the first round of reviews. I support the much clearer focus on the terminus types and geometry factors improved the paper substantially but of course the observed changes are not fully independent of climatic/oceanic forcing I somewhat agree with the re-reviewer that presenting no evidence of climatic/oceanic forcing and which makes it somewhat difficult to relate the observed changes to potential triggers (timing...). I leave it open to the authors whether they address this point but perhaps just showing for context some simple well established time series of atmospheric and oceanic (the latter being already tricky though) temperatures maybe useful.

We appreciate the concerns of reviewer 3 on the absence of climate forcing data from the manuscript. As the editor summarises, these data were included in the original submission, but we removed this to improve the focus on terminus types and geometry. We too feel that the paper was improved by removing it. We acknowledge that there are some points in the manuscript that we invoke climate forcing as a possible trigger of terminus retreat. However, we feel that these references to climate forcing are not extensive enough to warrant adding an assessment and time-series of climate-ocean forcing that would detract from the main focus of the manuscript. Some of the instances at which we inferred climate forcing have now been removed (i.e. when referring to Ryder Glaciers behaviour). Revisions have also been made to Section 4.1 following the specific comments to this section so that at all points at which we invoke climate forcing is a trigger of recent retreat. The referenced study also suggests that climate forcing may have triggered dynamic thinning. We feel that by making these changes we have alleviated some of the points at which referee 3 may have had concerns about the absence of climate/ocean forcing data.

**Referee** #5 Where is the calculation of the force balance, longitudinal stretching and driving stress that is referenced in the discussion Please make it explicit if we are supposed to deduce these from the velocity time series alone.

**Editor** #5 regarding the force balance I assume no explicit calculation has been done, if so rephrase this sentence somewhat to avoid confusion (if it has been done then explain it clearer).

We have now reworded this section to avoid confusion, following the specific line edit comment made by the referee.

**Referee** #6 Is the discussion of surge-type glaciers relevant to the main conclusions of the paper? It seems to confound the main points: behavior of grounded vs. floating termini and importance of bed topography controls.

**Editor** #6 I guess the fact that some surge-type glacier occur should be included but perhaps this can be shortened a bit to avoid loss of focus on the main important points.

We understand the concerns of referee 3 on the surge-glacier section. However we feel that it is worth briefly recording how surge-behaviour may cause glaciers to behave independently of terminus type. We have therefore followed the editor's suggestion to shorten this section of the discussion. We have done this by removing the previous fourth paragraph which provided a very brief (and thus potentially unnecessary) overview on glaciers that have been previously referred to as surge-type but are not found to be the case in this study. We have also made an effort to substantially shorten the paragraphs discussing likely surge-type glaciers. Another way we have improved this section is by providing better links to our main conclusions of the paper. Examples of this are on lines 31/7, 31/20-21, 32/19-25 in the tracked changes manuscript.

**Referee** #7 Perhaps most important --- the manuscript writing should be more clear and concise. The main point within individual sentences or paragraphs is often convoluted and, as a result, the content suffers significantly. I've tried to offer some specific improvements in my line edits, but was unable to address everything. Ultimately, these problems can be addressed with careful and collaborative editing by all authors.

**Editor** #7 please try to further make the writing in general more concise and clearer (simpler). Partly this will be solved by addressing the listed minor points by the editor. And further address list of minor points (which to some degree will make the manuscript more focussed and concise, see point #7

We have addressed the list of minor points made by the referee below, and feel that this has significantly improved the clarity of several parts of the manuscript. In each case we refer to line numbers in the tracked changes manuscript where changes have been made. We have also made an effort to make the writing more concise and clearer in several other places in the manuscript (e.g. Sections 3.2 and 3.3).

**Specific comments by page/line number and figure number:**

1/16. No need for parenthesis, just "was"

Deleted parenthesis.

1/19. "adjustment" not "re-adjustment"

Changed to adjustment.

1/21. Delete comma before suggests

Deleted.

1/29. Should be Carr, 2017a

We are unsure whether this comment is suggesting that we should be referencing Carr2017's paper on Novaya Zemlya, or if this is because it should be lettered a, based on the pan-arctic retreat paper being referred to first in the manuscript. If the former, 2017b (pan-arctic retreat) is a more appropriate reference for this sentence. If the latter,

the lettering has been done automatically by using The Cryosphere's referencing style. This bases the lettering on the time of year during the year that the paper was published. As this is the automatic referencing style, we assume we should just leave it as it is.

2/5. Delete "surface"

**Deleted.**

2/10. This paragraph is longwinded – considering stripping down to the main points, i.e., terminus retreat can initiate dynamic adjustments independent of climate and modulated by local outlet geometry and associated resistive stresses. The last two sentences seem most important.

We have made an effort to shorten this paragraph as suggested in an attempt to better highlight the main points (lines 2/8 to 2/22).

2/12. I suggest using "slow", "long", or "gradual", but best not to use two adjectives.

Changed to just 'slow periods of advance'.

2/17-19. Is this sentence necessary? If so, perhaps it should have a reference.

This sentence has now been removed when shortening this paragraph.

2/27. Delete "Most"

**Deleted.**

2/30. Create a new sentence..."Dynamic changes at Jakobshavn are linked to the gradual collapse of its floating ice tongue."

**Changed as suggested.**

2/31. Is there anything specific that can be added here to demonstrate the importance of northern Greenland ice dynamics to sea level rise? Important to let the reader know the region is important to study for reasons other than it's underrepresented in previous investigations.

We have added extra context to the importance of northern Greenland ice dynamics for sea level rise (lines 3/8-11).

2/33. Delete "far"

**Deleted.**

3/1. "Consequently, few long-term records of frontal positions exist in the region. As a result, their potential impact on inland ice flow remains unclear."

**Changed as suggested.**

3/5. The sentences in this paragraph seem redundant. I would suggest combining sentences 2-5 into something like, "We couple a multi-decadal annual terminus position record between 1948 and 2015 with recently published surface elevation and ice velocity datasets. We use these datasets to evaluate dynamic responses (i.e. acceleration and thinning) to frontal position change and examine disparities in the context of glaciers with floating or grounded termini."

**We have included this as suggested which has shortened and improved the clarity of this paragraph.**

3/10. Would recommend changing slightly to, "Finally, we assess local topographic setting (ie fjord width and depth) as a control on glacier behavior."

**Changed as suggested.**

3/16. Is this true? There are other, albeit smaller floating tongues elsewhere, such as Rink Isbrae and Helheim?

**To clarify we have changed 'extant' to 'long'.**

3/13-20. This seems like introduction or nonessential methods material. What is the point of this paragraph? Seems like most important information is the characterization of floating vs grounded termini... then quickly note that there are large and changing tongue systems.

This study area section was added in the last round of revisions after it was suggested by one of the reviewers. Their idea was that it would be good to have some background on the region and which glaciers still have floating ice tongues or are grounded across the region. We would like to at least introduce the region we are covering, but realise we can remove some of the excess detail to focus on the characterization of floating vs grounded termini. As such we have simply removed sentences 2 and 3 to shorten and remove nonessential material.

**5/2. Are there any gap years?**

We are unsure exactly what this point means. There are ortho-photos available for 1978 for most glaciers (excl. NW Greenland) and 1985 for glaciers in NW Greenland, so in some sense there are gaps, but it doesn't seem necessary to mention this.

6/7. To what end? Do you use changepoint analysis between glaciers, over a single record, etc.? What is the point? This paragraph needs a topic sentence that makes this clear up front for the reader to understand the value in this approach.

**We have now rewritten the first couple of sentences of this section (see lines 6/22 to 7/7).**

6/10. This sentence is redundant and could be more concise. Just cite Bunce and Carr in the first sentence after clarifying.

**See above.**

**6/18. Within what range?**

We have removed this sentence and just refer to the method used by Carr et al., 2017 to avoid confusion.

6/17-22. This explanation is confusing to the reader. What is the reason for a threshold penalty? What is a threshold penalty? If this method is following Carr 2017, then simply reference their method, give a brief overview with an emphasis on portraying what the main point is and why it's valuable. The main point seems to be articulated in the last sentence of the paragraph, perhaps this could be a topic sentence?

We have now explained more clearly what the threshold penalty value is, and how it assists in the automatic detection of the changepoints (7/11-18). While the method we use here is similar to that used by Carr et al., 2017, it does have some differences (i.e. our approach uses a function in Matlab rather than R), so we feel that some of the detail needs to remain. We have also added a topic sentence to this section which gives a better overview on the main point of using this method.

8/9. It is unclear why Euclidean distance is necessary - to draw centerlines? What if fjord walls are not parallel?

We have added some clarity to this section of the methods. In all cases the fjord walls were parallel, but we're not sure if there would be an issue if not, because this would still produce a maximum distance within the centre of any two lines. We still feel this method is more robust than drawing a centreline freehand.

8/11. I would think averaging elevation change over the entire centerline (to the ice divide as the manuscript suggests) would significantly skew your results. Would it also be better to mask elevation changes seaward of the grounding line on floating tongues?

We acknowledge that using 'to the ice divide is misleading' as the centreline profiles stop at the point at which the glaciers are no longer within a fjord rather than much further inland. When we mask out elevation changes seaward of the grounding line, the averaged elevation change values along the time series are the same in most cases. This suggests that including these data does not significantly skew the results. However, at a few glaciers by masking out changes along the tongue, we omit a lot of the data points from the earlier years of the elevation change record, when the data is generally sparser. For this reason we would like to keep the continuous trend averaged along the centreline. However, we do change the terminology in the methods to 'to the inland end of the glacier fjord' instead of the to the ice divide.

9/4. This sentence is unclear – you're calculating catchment areas from the flow field right? Could you instead reorient the sentence as, "We calculated each drainage area using catchments constrained by gradients in the DEM"...?

Yes we have calculated catchment areas based on flow direction and flow accumulation raster's derived from the gradients in the GIMP surface DEM. We have adjusted the final sentence of this paragraph as you have suggested.

9/7. Perhaps "Net retreat"?

Changed to 'net retreat'.

9/7. Do these statements pertain to frontal positions (ice tongue fronts and grounded termini), or just grounding lines? Please clarify.

Clarified by changing from 'glacier retreat rates' to 'frontal retreat rates'.

9/14. Is "mean rate of terminus change" more accurate?

Changed as suggested.

9/17. Could you be more direct and just say, "Long-term retreat rates varied across northern Greenland?"

Changed as suggested.

12/2-6. The distinction of terminus type needs to be made earlier to give the reader context to interpret records of terminus front change.

We have addressed this comment largely in our response to the major comment #2. We have restructured this section of the results following the advice given (lines 10/19 to 13/6).

12/12. Already stated previously. Need better topic sentence; why do grounded termini matter? State main result up front and then support with observations.

We have added a topic sentence to highlight the important contribution that calving from grounded outlets make towards ice discharge and sea level rise. We have also added a sentence summarising the main results which is that grounded outlet glaciers had substantially higher retreat rates during the last two decades of the study period (lines 14/4-6).

12/23. Already stated previously. Need better topic sentence; why do floating termini matter? State main result up front and then support with observations.

As above we have followed the suggestion to provide a better topic sentence and main summary of the results. We have now added sentences to the start of this section to this effect (see lines 14/18-20).

17/16. Higher with respect to what? Need to clarify.

Changed to 'had higher thinning rates than grounded termini...'

18/1, Perhaps change to, "different pattern of elevation change compared to the rest of the region: Storstrømmen and L. Bistrup Brae."

Changed as suggested.

19/1. Perhaps it is best to also explicitly separate this section into grounded vs. floating termini?

We have now added subheadings to this section to split it by grounded and floating termini.

19/4. Please clarify what is meant by "split"

To improve the clarity of this sentence we have now changed it to 'Some glaciers rest on inland sloping bed topography, while others have seaward sloping topography'.

19/2 and 22/1. What are the main points of these paragraphs? Please upgrade the topic sentences to better reflect the main point – inland sloping beds are correlated with higher retreat rates at glaciers with grounded termini, but fjord width is not a main determinant. What is a corresponding main point you are trying to get across in describing bed data at glaciers with floating tongues?

We are grateful for the suggestions to improve this section. We have followed the advice for each paragraph, by including a topic sentence followed by more focused key points for the bed topography based on each category of terminus type. In doing so we have also shorten these sections. We also make it clear why we are examining bed topography beneath the grounded portion of glaciers with floating ice tongues.

22/30. Delete acceleration and retreat in southeast Greenland, or support with references and include other regions where this occurred (e.g. SW Greenland, Howat et al., etc.).

As this sentence already contains a lot of information we have just deleted 'acceleration and retreat in south-east Greenland.'

22/31. Please provide more complete and recent references (e.g. Felikson et al., 2017). Also please more explicitly link these changes to climate factors, if that is in fact what you mean to do.

We have included this and other appropriate references (Price et al., 2011 and Nick et al., 2009) for the links between thinning and terminus change in this section.

23/2. Do you show this relationship in the text? If so cite a figure. Also please reword the sentence. "Indeed, surface thinning preceded rapid terminus retreat at many northern Greenland glaciers (Fig. x)." I would be wary of adding sentences referencing old studies in other parts of Greenland and focus more on the region of interest. What is the main point you are trying to make? Is it that climate may be the initial trigger for terminus change? Then make it explicit and cut the fat.

This was actually in reference to examples from other regions of the ice sheet, and we have now removed this sentence. We have made an effort to condense this section and make the key point that 'climate may be the initial trigger for terminus change', hopefully a lot clearer. We state this main point earlier in the paragraph. Sentences referencing old studies in other parts of Greenland have been removed and instead we have made better reference to previous studies in northern Greenland that have linked climatic changes with thinning and retreat. Finally we include the sentence suggested above as support for 'surface thinning being the driver of accelerated retreat since the 1990s'.

23/2. Confusing sentence, please reword. Not initial condition, do you mean forcing?

Changed to forcing

23/9. Delete "the"

Deleted.

23/8. I'm confused, is it climate and ocean forcing at the terminus; or thinning from negative mass balance that causes retreat? I realize they are related (i.e. climate driven), but I think you can be more direct.

In combining and restricting this section of the discussion this sentence has now been removed. Earlier in the section, we have made it clear that we are discussing dynamically induced thinning at the margins of outlet glacier termini, which may have been forced by changes in climate and ocean conditions at the margins.

23/12. This seems like material that could be combined with the previous paragraph. In fact, I would consider making this section a single paragraph that is more direct and punchier. Lots of material is repeated and the topic lacks focus.

We have outlined the changes we have made to this section in response to a previous point (23/2). We have now combined material that was previously in the second paragraph with the first and also made an attempt to shorten and summarise this section. In doing so we hope to have reduced the amount of material that is repeated and improved the focus of this section overall.

23/15. Can also add: Catania G. A., Stearns, L. A., Sutherland, D. A., Fried, M. J., Bartholomaus, T. C., Morlighem, M., et al. (2018). Geometric controls on tidewater glacier retreat in central western Greenland. Journal of Geophysical Research: Earth Surface, 123, 1–14, https://doi.org/10.1029/2017JF004499

Reference added.

23/25. What are the significant differences?

We have deleted this sentence as we did not deem it necessary, and the significant differences in the duration and magnitude of rapid retreat based on terminus type were very similar to the summaries of the 'calving behaviour based on terminus type' given in the previous sentence.

23/26. The last sentence of this paragraph is essentially the same as the first. Consider revising.

We have removed this sentence and revised this section. We now start with the first sentence that states that our analysis showed differences in glacier dynamic response to terminus change based on terminus type before going on to summarise the different calving behaviour based on terminus type.

23/31. What causes the initial near-terminus thinning? Is this supported in the results section? Does retreat lag thinning? If so, reference the appropriate figure. Otherwise, I feel this is unfounded.

We agree with this comment and major comment #3 (see additional response to #3) that it is very difficult and not well supported for us to make statements about near-terminus thinning, and especially the cause of this thinning given the absence of a detailed climate/ocean forcing analysis. Thus, we have shortened this paragraph and merged it with the first paragraph. We now hope to have better focused this section on the dynamic changes that occurred after terminus retreat began, rather than too much focus on the specific initial cause of retreat that we are unable to make from the data we use.

23/33. Ice does not flow inland; perhaps you mean the dynamic response propagates inland, or up-glacier?

Changed to 'and propagates the dynamic response (i.e. acceleration) up-glacier.'

23/33. Change to, "As such, we suggest thinning initiated enhanced retreat..."

Changed as suggested.

24/19. Awkward sentence, consider rewording.

This sentence has been revised in response the next comment.

24/19-20. Where do you show calculation of the force balance, longitudinal stretching and driving stress? Also, you may not have the temporal resolution in the velocity dataset to resolve short-lived dynamic adjustments from individual calving events.

See response to major comment #5. We do not calculate the force balance ourselves and instead infer this to be the case based on annual ice velocities. It is true that by looking at annual ice velocities only we may not be able to resolve short-lived dynamic adjustment. However the point we are trying to make here is that while there may have been some short-lived dynamic adjustments (seasonally), there was not a substantial perturbation in stresses that caused

these glaciers to accelerate over longer timescales (annually). To alleviate the confusion to the reader and improve clarity, we have amended these sentences on lines 28/15-20 in the tracked manuscript.

24/27. Delete the comma in this sentence.

**Deleted.**

28/10. It doesn't really seem that any glaciers are responding linearly to climate forcing. It looks like Ryder has a very nice, episodic advance/retreat cycle.

To clarify this and also address the comment below, we have instead changed this to 'appears to be behaving dissimilarly to the rest of our study glaciers in northern Greenland'.

28/10. It is hard to compare these records to climate forcing without also seeing time series of climate forcing (i.e. air or ocean temperatures).

We agree that it is hard to compare these records to climate forcing, but we feel that re-introducing such records would confuse the main message of the paper (as suggested by previous reviewers for removing it). To reduce the confusion in this example when revising this section we have simply removed the statement 'independently to climate forcing' and instead say that Ryder Glacier appears to be behaving 'dissimilarly to the rest of our study glaciers in northern Greenland'.

29/9. But didn't the results section showed fjord width had little control over retreat rates?

We have removed the word 'widening' to avoid confusion here, and just say 'Continuous retreat into deep fjords...'

Figure 6 and 7. It's hard to read and interpret yellow colored velocity data. I suggest changing to blue or another easily accessible color.

We have changed the colour of the velocity data in both figures to purple. This is hopefully now easier for the reader to interpret, but is less likely to be confused with the green surface elevation change than blue might be.

Figure 9 and 10. The bathymetry colorbars are not needed for each panel. Consider including one at the top or bottom of the figure.

We have removed the colour bars for each panel and left one colour bar for each figure (9 and 10).

Figure 11. Please annotate the ice flow direction and make explicit the location of ice tongues and lateral rifting. It's hard to make out these features without descriptions in the figure.

We have annotated the figure to show the ice flow direction, the location of the ice tongue and the locations of lateral rifting.

[revised manuscript text omitted]

---

## Author Response (AR3)

Dear Editor,

We are grateful for the acceptance of our manuscript for publication. Again we thank the Editor Andreas Vieli for the handling of our manuscript, and the three anonymous reviewers whose comments greatly improved the manuscript. We address the final minor points before publication below.

Best wishes,

Emily Hill

**Minor points to be considered:**

p. 2 line 33 and p. 12 line 4: at both instances the term 'dynamic ice discharge' is used which I find awkward (and I have not seen before) as 'discharge' already includes 'dynamic', maybe 'dynamic mass loss' is more appropriate
In both cases we have amended this to 'dynamic mass loss.'

p.18 line 12: is 'correlated' here really the appropriate term. Correlation mostly refers to a statistical relationship but no such statistical correlation has been undertaken. Maybe 'associated' would be more appropriate.
A similar issue occurs on p. 22 line 12.
In both cases we have amended this to 'associated.'

p. 22 line 27: there is an 'in' missing: '… The switch to terminus retreat IN the 1990s…'
Added in the missing 'in.'

p. 23 line 6: '…the initial forcing that accelerated…', I would not use the term 'forcing' here as the thinning is rather the initial PROCESS that causes the acceleration. Forcing wold rather be climate or ocean…
Changed 'forcing' to 'process'

p. 23 lines 14-15: maybe I am picky here but strictly speaking your data and analysis does not say anything directly about calving type, so saying '… we OBSERVE two dominant calving behaviours…' is not correct. You can only indirectly INFER anything on calving so I would replace 'observe' by 'infer'.
Changed 'observe' to 'infer'

p. 24 line 8: there is an 'of' missing: '…due to the calving OF large tabular…'
Added in the missing 'of'.

p. 24 line 16: '…substantial PARTS OF floating ice'
Added in 'parts of' before 'floating ice'

p. 27 line 19: awkward formulation '..too..also…' one of the two is probably enough.
Removed 'also'

[revised manuscript text omitted]